# Strong repulsive Lifshitz-van der Waals forces on suspended graphene

Gianluca Vagli [1,6], Tian Tian [1,2,6], Franzisca Naef[1], Hiroaki Jinno [1,5], Kemal Celebi[1], Elton J. G. Santos [3,4] & Chih-Jen Shih [1] ✉

Understanding surface forces of two-dimensional (2D) materials is of fundamental importance as they govern molecular dynamics in nanoscale proximity. Despite recent understanding of substrate-supported 2D monolayers, the intrinsic surface properties of 2D materials remain vague. Here we report on a repulsive Lifshitz-van der Waals force generated in proximity to the surface of suspended graphene. In combination with our theoretical model taking into account the flexibility of graphene, we directly measured repulsive forces using atomic force microscopy. An average repulsive force of up to 1.4 kN/m² has been detected at separations of 8.8 nm between a gold-coated tip and a sheet of suspended graphene, more than two orders of magnitude greater than the long-range Casimir-Lifshitz repulsion demonstrated in fluids. Our findings imply that suspended 2D materials could exert repulsive forces on any approaching electroneutral object in close proximity, resulting in substantially lower wettability. This could offer technological opportunities such as molecular actuation and quantum levitation.

When two electroneutral objects are brought in proximity in a polarizable medium, the correlations in their temporal electromagnetic (EM) fluctuations usually lead to an attractive interaction[1]. At small separations (<10 nm), this interaction is known as the vdW forces[2], while at large separations (>20 nm), where the retardation effect comes into play, it is termed the Casimir forces[3,4]. Early vdW theories[5-7] assumed the total interaction between two objects, each consisting of many molecules, is simply the sum of intermolecular potentials, which ignored the fact that the intermolecular interactions are strongly screened by the surroundings. By considering macroscopic properties using quantum field theory and statistical physics, seminal work presented by Lifshitz et al.[8] completely abandoned the pairwise additive assumption and predicted that quantum fluctuations can lead to repulsive interactions in both the vdW and Casimir regimes. The existence of Casimir repulsion was later experimentally verified in several fluid-based systems[9-13].

Consider two semi-infinite three-dimensional objects, A and B, interacting across a polarizable medium, m. As the interaction potential in the Lifshitz theory[2,8] is proportional to the product of effective polarizabilities of A and B screened by m, the most straightforward approach to generate Casimir or vdW repulsion is to design a set of materials such that[9,14,15]

$$(\varepsilon_A - \varepsilon_m)(\varepsilon_B - \varepsilon_m) < 0 \qquad (1)$$

where $\varepsilon_A$, $\varepsilon_B$, $\varepsilon_m$ are the frequency-dependent dielectric responses for A, B, and m, respectively.

Accordingly, the experiments demonstrating long-range Casimir repulsion were majorly carried out in high-refractive-index fluids, i.e., m = fluid, in which $\varepsilon_m$ is between $\varepsilon_A$ and $\varepsilon_B$ over a wide range of frequencies to obey inequality (1)[9,16-18].

[1]Institute for Chemical and Bioengineering, ETH Zürich, Zürich, Switzerland. [2]Department of Chemical and Materials Engineering, University of Alberta, Alberta, Canada. [3]Institute for Condensed Matter Physics and Complex Systems, School of Physics and Astronomy, The University of Edinburgh, Edinburgh, UK. [4]Higgs Centre for Theoretical Physics, The University of Edinburgh, Edinburgh, United Kingdom. [5]Present address: Institute of Space and Astronautical Science, JAXA, Chuo-ku, Sagamihara-shi, Japan. [6]These authors contributed equally: Gianluca Vagli, Tian Tian. ✉e-mail: chih-jen.shih@chem.ethz.ch

Unfortunately, the fluid dielectric response usually drops rapidly beyond the visible frequency region, lowering $\varepsilon_m$ below $\varepsilon_A$ and $\varepsilon_B$ that results in high-frequency attraction[19]. The long-range repulsive force observed in fluid arises from the retardation effect that diminishes the high-frequency contributions, but when working at small separations, the full-spectrum summation may convert the force from repulsion to attraction[20]. As the London dispersion remains an important part of the interactions, the measured Casimir-Lifshitz repulsion was rather weak (in the order of 0.1–10 N/m$^2$)[9].

Here we report on a stark repulsive Lifshitz vdW force generated in nanoscale proximity to the surface of a flat sheet of suspended graphene arising from its atomic thickness and dielectric birefringence properties. In combination with our simplified model, which accounts for the mechanical flexibility of graphene upon indentation, we directly measured the repulsive forces using atomic force microscopy (AFM). An average repulsive force of up to 1.4 kN/m$^2$ was detected at a separation of 8.8 nm between a gold-coated AFM tip and a sheet of suspended graphene, more than two orders of magnitude greater than the long-range Casimir-Lifshitz repulsion demonstrated in fluids[9]. Our findings imply that a flat sheet of suspended 2D materials could potentially exert repulsive forces on any approaching electroneutral object in close proximity, resulting in substantially reduced wettability, which we demonstrated through our evaporation experiments involving gold on suspended graphene. The enhanced Lifshitz-vdW repulsion could provide technological opportunities such as molecular actuation and controlled atomic assembly.

## Results

### Direct measurement of vdW repulsion

Recent findings of wetting transparency[21–25] and remote epitaxy on 2D material-coated substrates[26,27] represent an important inspiration for the analysis of the intrinsic surface properties of suspended 2D materials. Recently, experiments involving evaporation of gold on suspended graphene surfaces are particularly interesting as the gold (Au) deposits were found not to adhere to clean suspended graphene surfaces[28–30]. The substantially reduced wettability of suspended graphene motivated us to examine whether the vdW repulsion is the responsible mechanism. In order to examine our postulate, we first carried out direct measurements of surface forces on a sheet of suspended graphene using atomic force microscopy (AFM).

We transferred mechanically-exfoliated graphene onto two types of holey membranes, including (i) bare low-pressure chemical vapor deposition (LPCVD) grown silicon nitride (SiN$_x$), and (ii) 70 nm Au-coated SiN$_x$[31]. The latter type of holey membrane allowed us to electrically ground graphene during AFM measurement, thereby excluding the effects of electrostatic interactions. After graphene transfer, all samples were annealed in Ar/H$_2$ to remove contaminants[32,33]. The membrane hole where the suspended graphene was examined has a diameter of approximately 5 μm.

Two different gold-coated silicon nitride (SiN$_x$) tips with measured radii of 33 nm and 13 nm were chosen for the force-displacement measurements (see Supplementary Note 1 in Supplementary Material). A schematic diagram of the measurement system is shown in Fig. 1a. Upon AFM tip displacement, the experienced attractive or repulsive forces were recorded until establishing contact, which we define as the reference point corresponding to tip displacement, $d = 0$ nm, at minimum force in each measurement.

One noteworthy observation on the first measurements was that when establishing the contact, the force response is quadratic for freestanding graphene, in contrast to the linear response on supported regions (see Supplementary Figs. 4). This is well-known considering the mechanical flexibility of suspended graphene, which yields an elastic response of higher order[34]. Indeed, during the retraction process from a suspended graphene surface, the tip remains to adhere to graphene at a large tip displacement, revealing that both graphene

and the AFM cantilever were bent before breaking the physical contact. We also observed that the required force to break the physical contact is approximately equivalent for both suspended and supported graphene. We, therefore, infer that the dominant component of the contact mechanism during retraction is caused by capillary or meniscus forces[35,36], due to the condensation of water within the small gap between the tip and graphene surfaces. With the nonideality in mind, hereafter, we focus on the approach responses before physical contact with the sample surface. A plausible explanation to the contact mechanism upon approach will be elaborated later in this work.

Figure 1b–g presents representative topographical and force maps extracted based on 12321 and 5776 independent force-displacement measurements scanning over a $5.5 \times 5.5$ μm$^2$ area for monolayer graphene transferred on two types of holey membrane, (i) and (ii), respectively. In the latter set of measurements, we electrically grounded the graphene and AFM tip to eliminate any possible electrostatic interactions. The force maps (Fig. 1c, e) correspond to the surface force experienced by the AFM tip of 33 and 13 nm, respectively, at $d = 10$ nm.

The surface topography of suspended graphene varies from sample to sample due to the polymer-assisted transfer process that involves solvent drying. The non-uniform dissipation of liquid surface tension could result in some strain upon the graphene suspension. For example, the first sample exhibits some degree of surface corrugation close to the SiN$_x$ hole edge, with a larger flat domain located near the center (Fig. 1b). On the other hand, the overall topography of the second sample is more akin to a parabola (Fig. 1d).

Remarkably, for $d \geq 10$ nm, we clearly identified domains within both samples (Fig. 1c, e) where the AFM tip consistently experienced repulsive forces. The areas enclosed by white dashed lines correspond to the regions experiencing repulsive forces of $\geq 10$ pN. The average net repulsive force experienced by the smaller AFM tip in the second set of measurements (Fig. 1e) is approximately 40% of that in the first one, in which a larger tip was used (Fig. 1c). This observation is expected given the fact that the vdW force scales with the interacting area. Note that the repulsion observed here cannot result from charge interactions, since both samples, with and without electrical grounding, exhibit consistent behavior. Further comparison for the individual force-displacement responses characterized in both samples also suggested negligible influence of electrostatic interactions (Supplementary Fig. 5). Indeed, even there exists electrostatic interactions, since both graphene and gold are conductive, any charge trapped on the graphene surface will induce an image charge of opposite sign in gold that only leads to an attractive interaction.

In order to further reveal the correlation between the graphene surface landscape and the emergence of repulsive forces, Fig. 1f and g compare representative cross-sectional cuts through the repulsive domains combining the topographic and surface force profiles in two sets of measurements (Fig. 1d, e, respectively). We fitted the surface topography with a parabolic function $f_P(x)$ at a given point $x$. The force distribution allows us to deduce that the gold-coated AFM tip experienced repulsion when the graphene surface is relatively flat, with the maximum surface gradient, $\| \max\left(\frac{df_P}{dx}\right) \| \leq 0.01$, within the measurement domains considered here (for details see Section 2.3 in Supplementary Material).

We have carried out the same measurement protocols on both flat suspended and SiN$_x$-supported graphene using different AFM tips. Figure 2a–d compare 2D histograms for 144 force-displacement responses extracted from independent measurements of scanning force microscopy over an $600 \times 600$ nm$^2$ area of a sheet of flat suspended and SiN$_x$-supported graphene, using a gold-coated (Fig. 2a, c) and uncoated (Fig. 2b, d) SiN$_x$ tip of radius of 20 nm. The right panels present the force distributions at given $d$-cuts associated with the dashed lines in the left panels. For the measurements on suspended

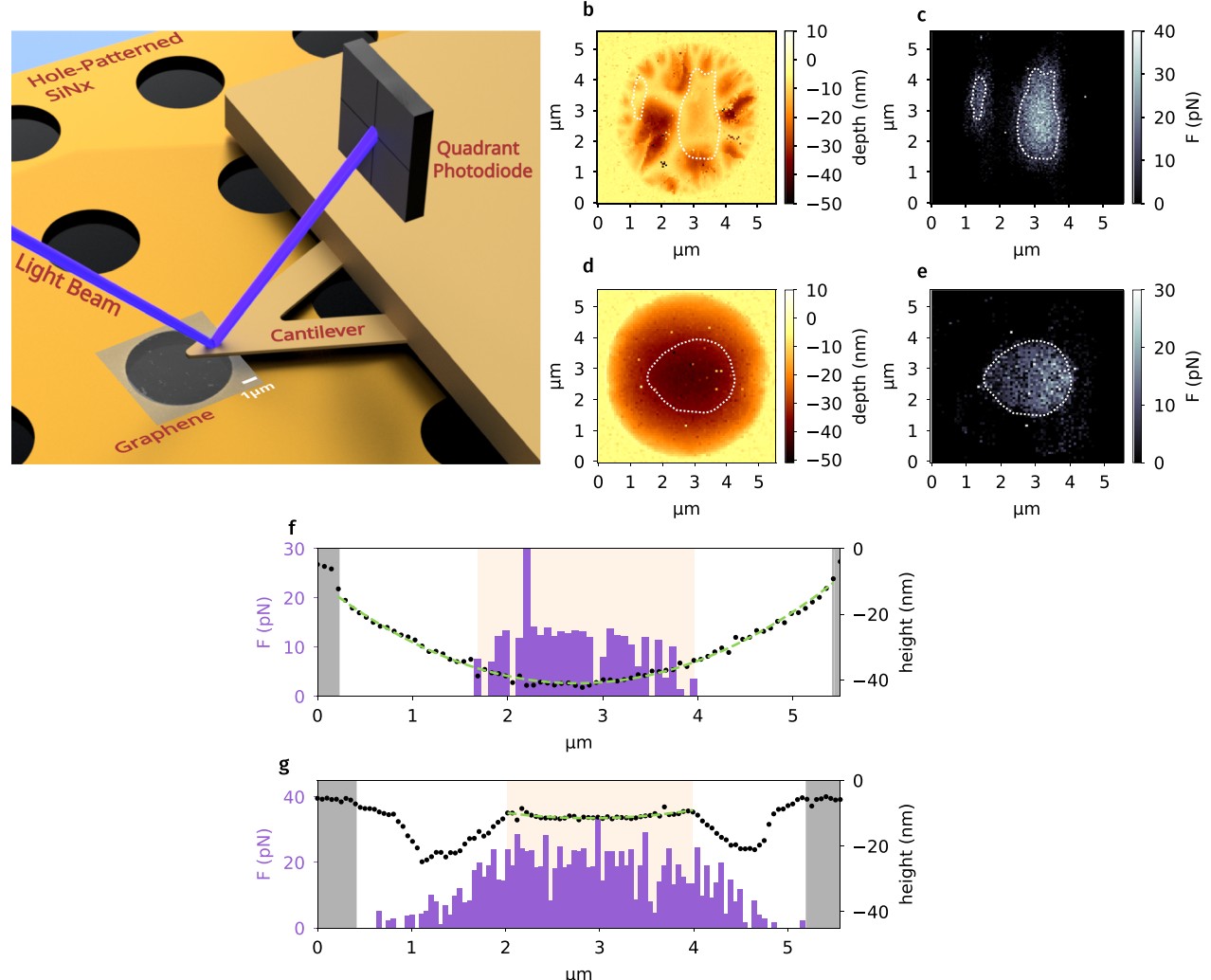

**Fig. 1 | Direct measurement of the Lifshitz-vdW repulsion on suspended graphene. a** Schematic diagram for the AFM measurement of the interaction forces experienced by a gold-coated AFM tip approaching graphene transferred to a holey membrane. **b, c** Representative topographical (**b**) and corresponding surface force (**c**) maps for a 33 nm radius gold-coated AFM tip interacting with a sheet of micromechanically exfoliated graphene transferred onto a LPCVD-grown SiN$_x$ holey membrane. The circular area of diameter of approximately 5 μm corresponds to suspended graphene. **d, e** Representative topographical (**d**) and surface force (**e**) maps generated using a 13 nm radius gold-coated AFM tip that approached another piece of graphene transferred onto an Au-coated SiN$_x$ holey membrane. The

surface force maps present the vertical force values experienced by the AFM tip at a displacement $d$ of ~10 nm before establishing the contact. The areas enclosed by white dashed lines correspond to the regions experiencing repulsive forces of ≥10 pN. **f, g** Representative cross-sectional cuts through the respective repulsive domains in (**d**) and (**e**), combining the topographic and surface force profiles. A parabolic function $f_P(x)$ was used to fit the topographic profile, suggesting that the repulsion emerges when graphene is relatively flat, with the maximum surface gradient, $\| \max\left(\frac{df_P}{dx}\right) \| \leq 0.01$, here highlighted with the light orange shading. The gray areas correspond to the boundaries of the SiN$_x$ pore.

graphene, both gold-coated and uncoated AFM tips started to experience repulsion from $d < 75$ nm, followed by a gradual increase with decreasing displacement. The last notable measured repulsive force before experiencing attraction was detected at an average displacement of 8.8 nm and 6.6 nm for gold and SiN$_x$ AFM tips, respectively. The corresponding Gaussian fits reveal a mean repulsive force of 11.8 ± 4.6 and 5.7 ± 4.0 pN for gold and SiN$_x$ tips experienced on suspended graphene, respectively. All of the repulsive force measurements presented here were proven to be of statistical significance (for details see Section 2.5 in Supplementary Material). On the other hand, only attractive responses were recorded on SiN$_x$-supported graphene (Fig. 2c, d) for both tips. The interaction appears to be relatively short-range, nearly negligible for large separations above 40 nm. The gold-coated and uncoated SiNx tips experienced weak attractive forces of − 0.6 pN ± 3.4 pN and − 2.0 ± 2.9 pN at $d = 18.5$ nm and $d = 36.0$ nm, respectively (Fig. 2c, d right), corresponding to the onset of attraction.

The results presented here also reveal that the measured interaction depends not only on the tip radius, but also on the tip material. Indeed, the measured repulsive force generated between a sharp Au-coated AFM tip (radius of 13 nm) and a flat suspended graphene reaches 7.5 ± 5.7 pN at $d \approx 7.5$ nm (see Supplementary Fig. 7), which is considerably greater than the counterpart for the less sharp SiN$_x$ AFM tip (radius of 20 nm), which yields repulsion of 5.7 ± 4.0 pN at $d \approx 6.6$ nm. Furthermore, in all measurements considered here, we observed a sudden emergence of an attractive force when the AFM tip displacement $d$ is smaller than approximately 5 nm. The observed transition to attractive force could be possibly attributed to the capillary interactions[36], although the scenario of attractive van der Waals (vdW) forces cannot be ruled out. In fact, the transition between attraction and repulsion has been observed and predicted for systems with complex geometries[37]. In the following section, we aim to describe the force-displacement responses by modeling the dielectric response of graphene and combining it with the Lifshitz theory of vdW forces.

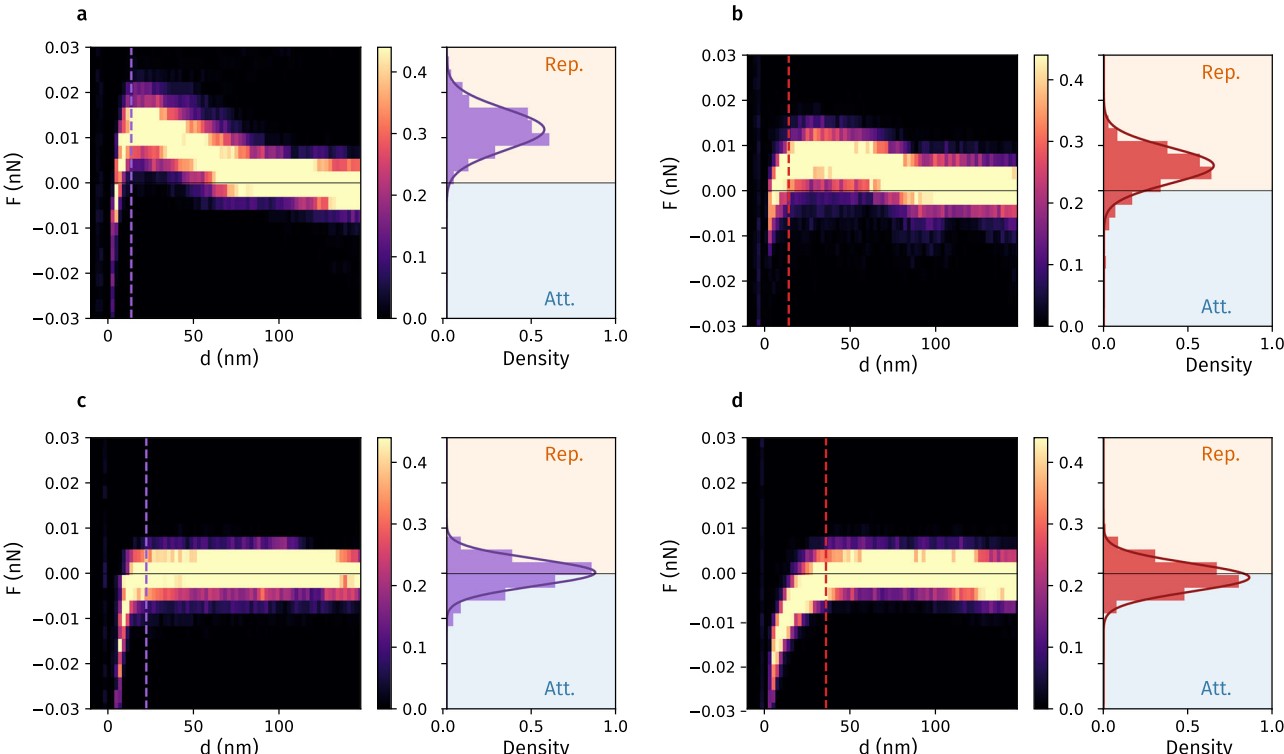

**Fig. 2 | Comparison of Lifshitz-vdW force-displacement responses in different systems.** 2D histograms of force-displacement responses for a gold-coated and an uncoated SiN$_x$ AFM tip approaching (**a**), (**b**) flat suspended graphene and (**c**), (**d**) SiN$_x$-supported graphene, respectively. The dashed lines represent the last notable measured force before experiencing attraction. For (**a**) and (**b**), this corresponds to a repulsive force detected at an average displacement of 8.8 nm and 6.6 nm, respectively.

## Modeling dielectric response of suspended graphene

In the past two decades, extensive theoretical effort has been devoted to understanding dispersion interactions mediated by 2D materials, with particular emphasis on graphene. Most studies are derived from the Lifshitz formalism, where graphene's dielectric properties are implicitly embedded within the Fresnel reflection coefficients. A notable contribution to this field, authored by Drosdoff and Woods, established an iterative formalism to calculate the Casimir interaction of graphene sheets[38]. Another seminal work comes from Klimchitskaya, Mostepanenko, and Sernelius, who proposed two methods to estimate graphene's dielectric properties, including (i) the density-density correlation function and (ii) the polarization tensor approach[39]. The latter was employed to examine the Casimir interaction of a gold-coated microsphere with SiO$_2$-supported graphene at large separations down to 224 nm[40]. However, none of these theoretical frameworks predicts the emergence of vdW repulsion between gold and suspended graphene at small separations, as experimentally observed here. We, therefore, suspected that the theoretical configurations considered in these studies may not adequately capture the nuance of the suspended graphene system.

Indeed, in the systems of suspended graphene[34], it is well-recognized that the 2D monolayer membrane exhibits extreme flexibility. The mechanical property is significantly different from that of a static interface, which is the basic assumption in the theoretical models discussed earlier. Furthermore, the interacting picture in our indentation measurement cannot be simplified as the approach of two semi-infinite plates, given the fact that the separation and the AFM tip radius are in the same order of magnitude. We notice that once the interacting configuration deviates from the planar geometries, a number of theoretical studies have suggested the possibility of repulsive forces under special circumstances within the Casimir regime. For example, the Casimir-

Polder interactions predict the generation of repulsion when graphene's temperature is significantly lower than that of the interacting metallic nanoparticle[41]. In the literature, there are also theoretical predictions[42,43] demonstrating the geometry-induced repulsive Casimir interactions, when the interacting configuration deviates from planar geometries. There are two particularly relevant reports involving an interlocking geometry[37,44] and a plane-sphere geometry of perfect conductors[45], which both closely resemble the indentation system considered here.

Although the geometry-induced Lifshitz-van der Waals forces are intriguing, they are particularly challenging to model considering the dynamic configuration. Inspired by the experimental findings of high bending radii of up to 1 nm for few-layer graphene[46] and the theoretical treatment by Milton et al.[14], where a medium of high permittivity intervening a low-permittivity space yields repulsion, we adopted an effective medium approach. Specifically, in this model, flexible graphene and vacuum are treated as a single, unified medium, as illustrated in Fig. 3a. The effective medium intervening between graphene and the surrounding vacuum upon the indentation of the AFM tip behaves analogously to a fluid immersion system. We consider that the suspended graphene surface follows a similar notion, being pulled into the space between itself and the AFM tip upon approach.

With the rationale of our effective medium approach in mind, the system may be simplified to a one-dimensional Lifshitz formalism in which gold (B) interacts with vacuum (A) across an effective medium containing monolayer graphene (m) (Fig. 3b). We treat the effective medium m as a birefringent space with in-plane (IP) and out-of-plane (OP) dielectric responses $\varepsilon_m^{\parallel}$ and $\varepsilon_m^{\perp}$, which are functions of separation $z$ and imaginary frequency $i\xi$, resulting from distinct IP and OP electronic properties of graphene. Indeed, from a dielectric screening point of view, recent findings according to the density functional theory (DFT)

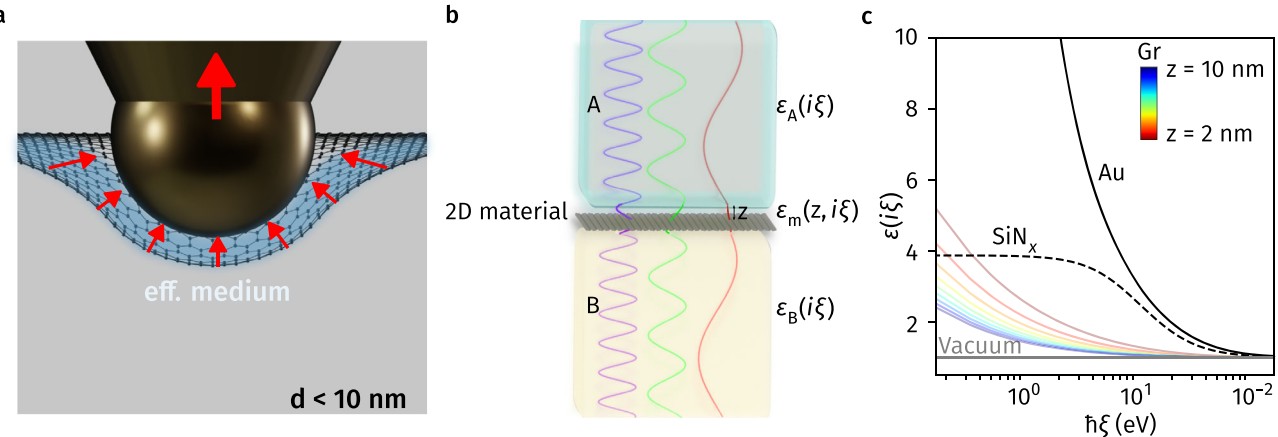

**Fig. 3 | Lifshitz-vdW interaction at surfaces of suspended 2D materials.** Schematic illustration of (**a**) the assumption of treating graphene and the intervening space as an effective fluid immersion medium for small separations, where graphene is significantly distorted upon the indentation of an AFM tip. **b** Schematic representation of Lifshitz formalism taking into account of the effective medium assumption. The interaction potential between materials A and B across a birefringent medium gap m containing a sheet of monolayer 2D material becomes repulsive when $\left[\varepsilon_A(i\xi) - \hat{\varepsilon}_m(i\xi)\right]\left[\varepsilon_B(i\xi) - \hat{\varepsilon}_m(i\xi)\right] < 0$. **c** Dielectric responses for Au, SiN$_x$, Vacuum and Gr as a function of electromagnetic energy, $\hbar\xi$, for different separations. Accordingly, we suggest that Lifshitz-vdW repulsion may be observed for A/m/B = Vac/Gr/Au at any separation for all frequencies.

calculations have suggested that the dielectric response for a sheet of suspended graphene is highly influenced by the size of the surrounding vacuum[47]. In order to properly model the birefringence for the medium taking into account the surrounding vacuum, based on recent Lifshitz formalism for the calculation of vdW interactions of layered material[47,48], it follows $\varepsilon_m^{\parallel}$ and $\varepsilon_m^{\perp}$ are given by $\varepsilon_m^{\parallel}(z) = 1 + \frac{\alpha_{2D}^{\parallel}}{\varepsilon_0 z}$ and $\varepsilon_m^{\perp}(z) = \left(1 - \frac{\alpha_{2D}^{\perp}}{\varepsilon_0 z}\right)^{-1}$, respectively, where $\alpha_{2D}^{\parallel}$ and $\alpha_{2D}^{\perp}$ are the $z$-independent IP and OP polarizabilities for the 2D material extracted from first-principles calculation (see "Methods" for details).

Figure 3c compares the dielectric responses for gold (Au), silicon nitride (SiN$_x$), vacuum (Vac), and graphene (Gr) for different separations $z$. For graphene's dielectric responses, the geometrically-averaged dielectric functions $\hat{\varepsilon}_m = \sqrt{\varepsilon_m^{\parallel} \varepsilon_m^{\perp}}$ are used here. Our calculations suggest that the vdW repulsive forces may be generated in two sets of material systems: (i) A/m/B = Vac/Gr/Au and (ii) A/m/B = Vac/Gr/SiNx. In particular, the former obeys the dielectric mismatch condition in inequality (1) in all separations and frequencies, yielding full-spectrum repulsion, under the assumption that the effective medium approach is valid.

We note that the earliest demonstration of Lifshitz-vdW repulsion shared a similar scenario, in which the repulsion generated between the container wall (B) and vacuum (A) through a superfluid helium film (m) resulted in fluid climbing[49]. However, in our case, the replacement of liquid films with 2D monolayers not only enables significantly wider spectral coverage, but also permits sub-10-nanometer separations, which potentially enhance the interaction, as the energy defined by the Lifshitz formalism scales with the inverse-square law within the vdW regime[2,50,51].

## Data evaluation with effective medium model

We have implemented the theoretical framework based on the effective medium approach to evaluate the measured force-displacement responses on the surface of suspended graphene. Specifically, the vdW interaction between two bulk materials A and B per unit area, across a birefringent medium m, $\Phi_{AmB}^{vdW}$, as a function of separation $z$, is given by ref. 2:

$$\Phi_{AmB}^{vdW}(z) = \sum_{n=-\infty}^{\infty} \frac{k_B T g_m(i\xi_n)}{16\pi z^2} \left\{ \int_{r_n}^{\infty} q \ln\left[1 - \Delta_{Am}(i\xi_n)\Delta_{Bm}(i\xi_n)e^{-q}\right] dq \right\}$$

(2)

where $k_B$ is the Boltzmann constant, $T$ is the absolute temperature, $\xi_n = 2\pi n k_B T/\hbar$ is the n-th Matsubara frequency, $\hbar$ is the reduced Planck constant, $r_n = \frac{2d\xi_n}{c}\sqrt{\varepsilon_m}$ is the retardation factor[2], $c$ is the speed of light in vacuum, and $q$ is a dimensionless auxiliary variable. $g_m = \varepsilon_m^{\perp}/\varepsilon_m^{\parallel}$ is the dielectric anisotropy[47] of m. Note that this approach was also used to calculate the vdW interactions of layered materials[52] to compute the dielectric responses of monolayers in vacuum, as we illustrated earlier. $\Delta_{Am}$ and $\Delta_{Bm}$ correspond to the dielectric mismatches following $\Delta_{jm} = \frac{\hat{\varepsilon}_j - \hat{\varepsilon}_m}{\hat{\varepsilon}_j + \hat{\varepsilon}_m}$, for j = A, B. Analogous to inequality (1), the vdW potential for a given EM mode $\xi_n$ becomes positive when $\Delta_{Am}\Delta_{Bm} < 0$, contributing to vdW repulsion. Accordingly, the vdW force per unit area generated between A and B, $F_{AmB}^{vdW}$, is given by[8,50,51]:

$$F_{AmB}^{vdW}(z) = \sum_{n=-\infty}^{\infty} \frac{k_B T g_m(i\xi_n)}{16\pi z^3} \left\{ \int_{r_n}^{\infty} q^2 \frac{\Delta_{Am}(i\xi_n)\Delta_{Bm}(i\xi_n)e^{-q}}{1 - \Delta_{Am}(i\xi_n)\Delta_{Bm}(i\xi_n)e^{-q}} dq \right\}$$

(3)

Although the dielectric response for a 2D material is $z$-dependent, namely $\hat{\varepsilon}_m(z, \xi_n)$, its partial derivative with respect to $z$ only contributes to corrections of higher order. Eq. (3) is sufficiently accurate to approximate the exact solution.

Figure 4 a presents the calculated $-F_{AmB}^{vdW}$, as a function of separation $z$ for A/m/B = Vac/Gr/Au, Vac/Gr/SiN$_x$, SiN$_x$/Gr/Au and SiN$_x$/Gr/SiN$_x$. We further compare the calculated response for A/m/B = SiO$_2$/Bromobenzene(BB)/Au, benchmarking the fluid immersion system that quantitatively demonstrates Lifshitz-Casimir repulsion at large separations[9]. The three systems showing repulsive responses, A/m/B = Vac/Gr/Au, Vac/Gr/SiN$_x$ and SiO$_2$/BB/Au, have similar strength for $z > 30$ nm. The liquid immersion system is even slightly larger for $z > 50$ nm (See Fig. 2 in Supplementary Material), because the dielectric response of the graphene effective medium drops rapidly with increasing separation. Nevertheless, as pointed out by Boström et al.[19], in the SiO$_2$/BB/Au system, the receding retardation effect turns the Casimir repulsion to vdW attraction for $z < 20$ nm, exhibiting a maximum repulsive force of $\sim 0.015$ kN/m$^2$ at $z \approx 25$ nm.

For the interactions with SiN$_x$-supported graphene, the calculated $-F_{AmB}^{vdW}(z)$, A/m/B = SiN$_x$/Gr/Au, SiN$_x$/Gr/SiN$_x$ nicely capture the features of the measured $F - d$ responses (Fig. 2c and d), in which the attractive force starts to emerge at $d$ (or separation $z$) of $\approx 30$ nm. In addition, the experimentally observed attractive response for SiN$_x$/Gr/

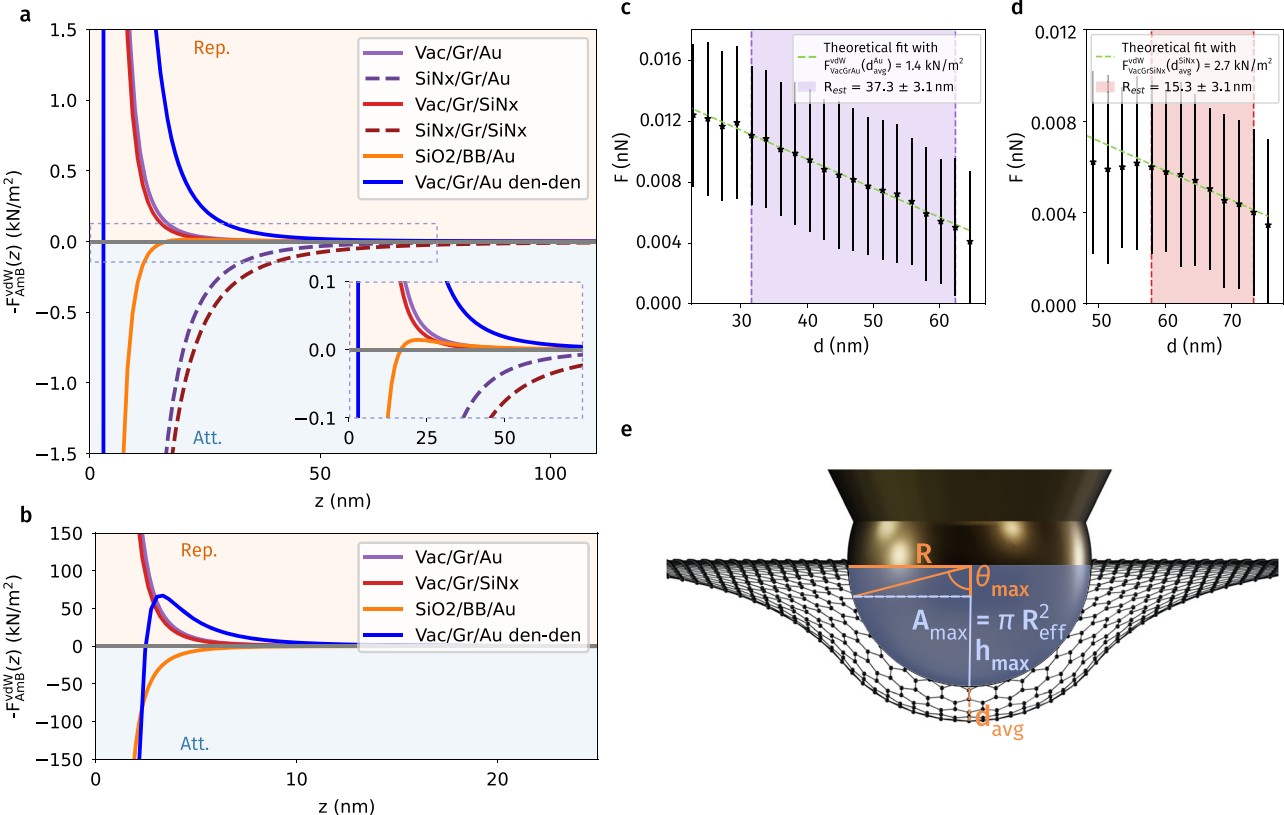

**Fig. 4 | Comparison of measured repulsive forces with Lifshitz theory taking into account the deformation of suspended graphene. a, b** Comparison of Lifshitz theory-calculated vdW forces per unit area as a function of separation $z$ using the effective medium approach for A/m/B = Vac/Gr/Au, Vac/Gr/SiN$_x$, SiN$_x$/Gr/Au, SiN$_x$/Gr/SiN$_x$ and SiO2/BB/Au and Vac/Gr/Au using the density-density (den-den) correlation function for the description of the polarizability from refs. 39,53 is also attached. **c, d** AFM-measured repulsive forces with the error bars being the

standard deviations of the distributions calculated from 144 measurements as a function of tip displacement $d$ for gold (**c**) and SiN$_x$ (**d**) AFM tips. In both cases, there exists a regime where the repulsive force linearly increases with $d$, where the tip displacement only deforms graphene without changing the average separation between tip and graphene, $d_{avg}$. **e** Schematic diagram showing the deformation of suspended graphene and the extracted geometric parameters at $d = d_{avg}$, yielding the maximum repulsive force detected between tip and graphene.

SiN$_x$ (Fig. 2d) is more long-ranged compared to SiN$_x$/Gr/Au (Fig. 2c), in coherence with the calculations shown in Fig. 4a.

The calculated Vac/Gr/Au and Vac/Gr/SiN$_x$ responses are monotonically repulsive for separations $z < 30$ nm. We also implemented the density-density (den-den) correlation function for the description of the polarizability suggested by Sernelius et al.[39,53] in our effective medium approach. The calculated response is shown in Fig. 4a and b (the solid blue curves), which, on the other hand, exhibits a sudden transition from repulsion to attraction at a separation of approximately $z \approx 2.1$ nm. The attraction appears to come from an increase in polarizability at small separations, resulting from a shift in the relationship between the wavevector and frequency relative to the Fermi velocity. At this point, we could not assert that the density-density correlation function-induced attraction explains the experimentally observed $F - d$ responses at small displacements on suspended graphene, as the dielectric response of graphene at small separations remains controversial. However, the attractiveness of this subnanometer regime is undeniable, as it was not only reported, but also extensively measured and analyzed in previous works by Chiou et al.[54] Considering prior literature and the limitations of our theoretical framework, hereafter, our discussion will focus on the calculations at intermediate displacements, $6$ nm $< d < 60$ nm.

Our calculated repulsion scales approximately with $z^{-3}$ and reach ~1.3 kN/m$^2$ and ~0.9 kN/m$^2$ at $z = 9$ nm for Vac/Gr/Au and Vac/Gr/SiN$_x$, respectively. The values calculated using the density-density correlation function are even higher, both being orders of magnitude higher than the fluid immersion system, SiO2/BB/Au. Nevertheless, we

realized that it is not proper to directly evaluate the AFM-measured $F - d$ responses on suspended graphene using the calculated $-F_{AmB}^{vdW}(z)$ profile. A major concern is that upon AFM indentation, the displacement of the AFM tip, $d$, is not equal to the separation between Au and Vac, $z$, owing to the mechanical deformation of suspended graphene. More specifically, when the AFM tip is approaching, the generated initial long-range attraction rather bends and deforms the suspended graphene before turning into a net repulsion, so that the tip displacement does not effectively represent the separation between the tip and graphene. The scenario involving the initial long-range attraction explains also why the repulsion was detected already at large displacements ($d \approx 75$ nm; see Fig. 2a, b), as compared to the theoretically suggested onset separation ($z \approx 20$ nm; see Fig. 4a).

We have developed an approach to evaluate our measurements which take into account the effect of deformation of suspended graphene. Fig. 4c and d present magnified $F - d$ responses together with standard deviations as a function of $d$ extracted from Fig. 2a, b. We found that there exist linear regimes, $25 < d < 62$ nm for gold (Fig. 4c) and $58 < d < 73$ nm for SiN$_x$ (Fig. 4d) tips, where the detected repulsive forces increase linearly by reducing the displacement. One could notice that the widths of these regimes are approximately equal to the corresponding AFM tip radii, 33 nm and 20 nm for gold and SiN$_x$, respectively. This behavior is consistently observed over different samples, including the electrically grounded sample shown in Fig. 1d.

A plausible picture informing the observed linear regime may share the scenario proposed by Santos et al.[55] who demonstrated a linear dependence between the critical free amplitude during AFM

indentation with respect to the tip radius. We inferred that the linear regime essentially measures the repulsive force experienced by the AFM tip during an indentation process that deforms graphene from a flat plane to a hemispherical surface, as revealed in Fig. 4c, d, without reducing the average separation between the tip and graphene, $d_{avg}$. One could readily estimate the tip radii, $R_{est}$, by characterizing the width of the linear regime, yielding values of $37.3 \pm 3.1$ nm, $15.3 \pm 3.1$ nm (Supplementary Fig. 7), and $13.2 \pm 3.1$ nm for the two gold-coated and bare $SiN_x$ tips, respectively.

More specifically, within the linear regime, consider the AFM tip of a hemispherical tip with radius of $R$ approaching a flat sheet of suspended graphene. When the repulsive force is sufficiently strong at a small separation, the increase of measured force with displacement primarily results from the increase of interacting area, $A$, upon indentation. Indeed, considering the facts of (i) relatively small dielectric response $\varepsilon$ of $SiN_x$ compared to Au (Fig. 3c) and (ii) similar cantilever stiffness, $k_{Au} = 0.16136$ N/m and $k_{SiN_x} = 0.11670$ N/m, it is reasonable to infer that the average separation required to bend the AFM cantilever for the $SiN_x$ tip is smaller than the gold counterpart, $d_{avg}^{SiN_x} < d_{avg}^{Au}$. With this in mind, given the significant deflection of repulsive force at an average displacement $d = 8.8$ nm for gold and $d = 6.6$ nm for $SiN_x$ before bouncing into contact, we let $d_{avg}^{Au} = 8.8$ nm and $d_{avg}^{SiN_x} = 6.6$ nm for proper comparison with our calculations. Following the scenario, beyond the linear regime, the repulsive force exhibited a plateau for $8.8 < d < 25$ nm and $6.6 < d < 58$ nm, for gold and $SiN_x$ tips, respectively, where the interacting area between AFM tip and deformed graphene remained nearly unchanged upon indentation, reaching its maximum $A_{max}$ (see Fig. 4e).

According to the physical picture presented above, considering the spherical geometry of the tip, we model the slope of the $F − d$ response in the linear regime, $m_{fit}$, following:

$$m_{fit} = − F_{AmB}^{vdW}(z = d_{avg})\pi R_{eff} \qquad (4)$$

where $R_{eff} \approx R_{est}$ is the effective radius of the projected circular area interacting with graphene (see Fig. 4c). Using the experimentally extracted value $m_{fit}^{Au} = − 0.20$ mN/m and $m_{fit}^{SiN_x} = − 0.13$ mN/m and the Lifshitz-theory calculated force density $F_{VacGrAu}^{vdW}(z = d_{avg}^{Au}) = 1.4$ kN/m$^2$ and $F_{VacGrSiN_x}^{vdW}(z = d_{avg}^{SiN_x}) = 2.7$ kN/m$^2$, the geometrical parameters $R_{eff}$ and $A_{max}$ at the deflection points $d_{avg}^{Au} = 8.8$ nm and $d_{avg}^{SiN_x} = 6.6$ nm (see Fig. 4e) are determined to be 43 nm and 5949 nm$^2$ for gold and 15 nm and 720 nm$^2$ for $SiN_x$, respectively. Although all the extracted parameters are quantitatively reasonable with the estimated radii $R_{est}$ and the separation range considered here, we believe that a more accurate description of the system is possible by further implementing the new theoretical analysis in references[39], which may be a promising candidate to elucidate the experimentally observed sudden emergence of attractive forces at small separations in Fig. 2.

## Nonwettability of suspended graphene

Our findings imply that a locally flat area on a sheet of suspended 2D monolayer may be intrinsically repulsive, hindering molecular adsorption and deposition that results in a lowered surface wettability. Indeed, the effective medium approach in combination with the Lifshitz formalism suggests that the effective dielectric response of a sheet of suspended 2D monolayer is only slightly higher than vacuum at sub-100 nm proximity. As a result, any electroneutral object approaching the surface could experience a repulsive force, smaller or larger, depending on the interacting configuration. More specifically, the key parameters include: (i) the materials permittivity, as it determines the product of $(\varepsilon_A − \varepsilon_m)(\varepsilon_B − \varepsilon_m)$ and (ii) the separation and object geometry (or the effective interacting area), which directly scale the interactions.

We examined the postulate by depositing gold particles on suspended graphene through an electron-beam evaporation process in

high vacuum. The evaporation source generated high-energy gold vapor particles, which then condensed on a sheet of exfoliated graphene transferred onto a porous $SiN_x$ window. We deposited a small amount of gold ($\approx 0.1$ nm thickness) at a rate of 0.01 nm/s on the sample surface at room temperature. As revealed in the SEM images of Fig. 5 for graphene with different numbers of layers, one can clearly identify two regions, namely the $SiN_x$-supported and suspended graphene, which exhibit distinct morphology and density for the deposited gold clusters.

On $SiN_x$-supported graphene, due to the very high surface energy of gold, fast condensation at room temperature yields small nanoclusters with high nucleation density. However, on suspended graphene, despite a high degree of supercooling, the nucleation density is extremely low with a preference for heterogeneous nucleation sites.

In particular, Fig. 5 directly compares the nucleation behavior of gold particles grown on suspended mono- (Fig. 5a and Fig. 5b) and 4-layer (Fig. 5c and d) graphene. To reach statistical significance, for each graphene layer, histograms of gold particle sizes were extracted based on three independent areas of $SiN_x$ holes on the same sample of exfoliated graphene (Supplementary Fig. 10). On the monolayer graphene areas, we noticed that most nucleation sites took place along straight lines, hypothetically corresponding to the ripples induced during the annealing and transfer process, as suggested in ref. 30. On the contrary, the nucleation sites on the 4-layer areas distribute more randomly, suggesting that the 4-layer areas are intrinsically flatter due to its relative mechanical stability. Even when we count all particles accumulated along the ripples, within a circular suspended area of 5 μm diameter, our analysis reveals average counts of nucleation sites of $586 \pm 52$ and $906 \pm 169$ for mono- and 4-layer graphene, respectively. The nucleation density of gold particles on suspended monolayer graphene is approximately 35% lower than that on the 4-layer counterpart. These findings can be nicely explained by the generation of Lifshitz-vdW repulsion on suspended graphene, which, according to our calculations (Supplementary Fig. 12), becomes stronger by reducing the graphene thickness. The amplification of Lifshitz-vdW repulsion on suspended monolayer graphene substantially lowers its wettability.

We noticed that our observation of increased nucleation sites on thicker suspended graphene exhibits an opposite trend compared to the nucleation behavior on $SiO_2$-supported graphene[56,57]. Indeed, as revealed in Eq. (1), the presence of an underlying substrate, such as $SiO_2$, could convert the Lifshitz-vdW interaction from repulsion to attraction, thus affecting the nucleation behavior of gold. A recent report by Frances et al.[30] also observed the aggregation of gold particles on the ripples of suspended monolayer graphene, but the nucleation density remained to decrease with the layer number of suspended graphene. We suspect that the deposition rate, or the kinetic energy of the incident gold particles, could also play an important role. More detailed and systematic experimentation is required to fully understand the phenomenon. Nevertheless, we would like to point out that the extremely low surface wettability of suspended monolayer graphene observed here cannot result from atomic smoothness, as its mechanical flexibility typically induces a higher degree of surface corrugation during the transfer process.

## Discussion

We have directly observed the strong repulsive Lifshitz vdW forces generated on a flat sheet of suspended graphene using an AFM. We have developed a theoretical framework combining Lifshitz formalism and the effective medium approach to describe the Lifshitz-vdW repulsion, which in general shows reasonable agreement. We would like to point out the limitation of our current model in a certain range of separations. The theoretical framework can be potentially improved by implementing, for example, the polarization tensor approach[39]. The

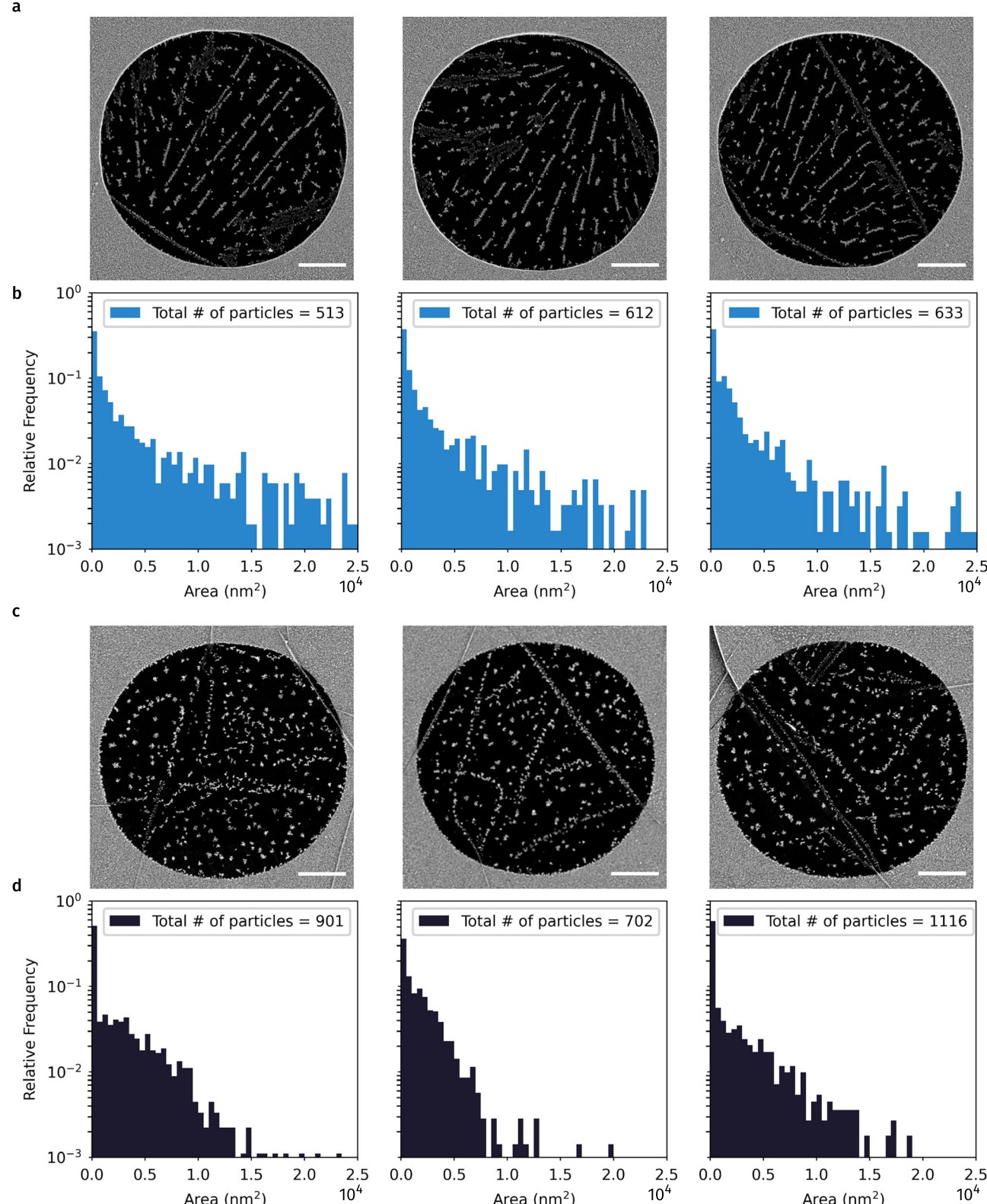

**Fig. 5 | Lowered wettability of suspended graphene that reduces the nucleation of gold particles.** SEM images and histograms of gold particle size comparing the nucleation behavior of suspended mono (**a**, **b**) and 4-layer (**c**, **d**) graphene areas in the same sample. The gold particles were deposited in an e-beam evaporator at 0.01 nm/s and a nominal thickness of 0.1 nm. The increase in the number of graphene layers enhances gold nucleation, which can be nicely explained by a reduced Lifshitz-vdW repulsion. Scale bars: 1 μm.

scenario of Lifshitz-vdW repulsion can nicely explain the substantially lowered wettability of suspended graphene observed in the evaporation experiments. The generation of strong Lifshitz-vdW repulsion can be used to realize quantum levitation[9,11,58] without fluid immersion, which gives rise to new nanoelectromechanical systems. In general, we believe that the manipulation of surface forces and the processing of suspended 2D materials will be greatly facilitated by the fundamental insights presented here.

## Methods

### Calculation of vdW interaction spectra

The dielectric response as a function of imaginary frequency $\varepsilon(i\xi)$ is calculated using the Kramers-Kronig relationship[2] given by:

$$\varepsilon(i\xi) = 1 + \frac{2}{\pi} \int_0^\infty \frac{\omega \text{Im}[\varepsilon(\omega)]}{\omega^2 + \xi^2} d\omega \tag{5}$$

where $\omega$ is the real frequency, and $\text{Im}[\varepsilon(\omega)]$ is the imaginary part of the complex dielectric function $\varepsilon(\omega)$. Frequency-dependent dielectric functions of $SiO_2$[59], $SiN_x$[59], bromobenzene (BB)[9] and Au[59] were extracted from experimental data.

Frequency-dependent 2D polarizabilities ($\alpha_{2D}^{\parallel}$, $\alpha_{2D}^{\perp}$) of graphene were calculated using the projector augmented wave method[60] in the ab initio package GPAW[61]. Dielectric responses were calculated using random phase approximation on top of the Perdew–Burke–Ernzerhof exchange-correlation functional[62] with plane wave cutoff energy of 500 eV, k-point density of 15 Å$^{-1}$ and truncated Coulomb kernel to avoid spurious interaction from periodic images.

### Exfoliated graphene on silicon nitride chip

Graphene flakes were exfoliated[63] from natural graphite and transferred to a silicon nitride ($SiN_x$) chip of 1 cm$^2$ size, patterned with an $12 \times 12$ hole-matrix, in which both hole-diameter and separation distance is approximately 5 $\mu$m. The window with the hole matrix was fabricated using electron beam lithography and etching. A wedging transfer method[31] was chosen to avoid damaging the hole-patterned structure. During the transfer, the polymer-graphene structure was aligned with a micro manipulator to the hole matrix.

### Electron microscopy

SEM characterizations were carried out on Zeiss ULTRA plus with a 0.8 kV beam voltage and 20 µm aperture.

### AFM setup and calibration

The force-distance measurements were carried out with a Bruker BioScope Resolve AFM using the B side of two gold-coated NPG-10 and an uncoated DNP-10 Bruker ($SiN_x$) AFM tip, with estimated radii $R$ of approximately 33 nm, 13 nm and 20 nm, as highlighted with a red dashed circle in Supplementary Fig. 3a–c, respectively. In order to obtain the interaction force (N) from the original measured values of the piezo element displacement in (mV), we carried out tip calibration steps to extract the deflection sensitivity $\left(\frac{m}{V}\right)$ and spring constant $\left(\frac{N}{m}\right)$, by using the PeakForce™ QNM™ suite within the NanoScope®software environment. We determined deflection sensitivities of 69.679 $\frac{nm}{V}$, 86.484 $\frac{nm}{V}$ and spring constants of 0.16136 $\frac{N}{m}$, 0.14710 $\frac{N}{m}$ for both Au-coated tip, respectively and a deflection sensitivity of 75.567 $\frac{nm}{V}$ and a spring constant of 0.11670 $\frac{N}{m}$ for the uncoated tip. The calibrated spring constants are comparable to the nominal value of 0.12$\frac{N}{m}$ for both, provided by the tip vendor.

## Data availability

All presented source data generated in this study are provided in the Source Data file. Further data supporting the findings are available from the corresponding author upon request. Source data are provided in this paper.

## Code availability

The codes that support the findings of this study are available from the corresponding authors upon request.

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

## Acknowledgements

C.J.S. is grateful for financial support from ETH startup funding and the European Research Council Starting Grant (N849229 CQWLED). E.J.G.S. acknowledges computational resources through CIRRUS Tier-2 HPC Service (ec131 Cirrus Project) at EPCC, funded by the University of Edinburgh and EPSRC (EP/P020267/1); ARCHER UK National Supercomputing Service (http://www.archer.ac.uk) *via* Project d429. E.J.G.S. acknowledges the EPSRC Open Fellowship (EP/T021578/1) and the Edinburgh-Rice Strategic Collaboration Awards for funding support.

## Author contributions

G.V., T.T., and C.J.S. conceived the idea and designed the experiments. G.V., T.T., and F.N. developed the theoretical framework. T.T. performed first-principle calculations under the guidance of E.J.G.S. G.V. of carried out AFM force measurement, analyzed the data, and modeled the force responses with the help of H.J. G.V., T.T., and K.C. fabricated the free-standing graphene samples. G.V. and T.T. characterized the free-standing graphene samples. G.V., T.T., and C.J.S. co-wrote the paper. All authors contributed to this work, read the manuscript, discussed the results, and agreed to the contents of the manuscript and supplementary materials.

## Funding

## Competing interests

The authors declare no competing interests.
