## [Transparent Peer Review file · Nature Communications]

Strong Repulsive Lifshitz-van der Waals Forces on Suspended Graphene

Corresponding Author: Professor Chih-Jen Shih

Version 0:

Reviewer comments:

Reviewer #1

(Remarks to the Author)

This work presents theoretical calculations predicting vdW-repulsion in a gold/graphene/vacuum system. To support this finding, the authors perform AFM measurements of suspended graphene using a gold-coated AFM tip and determine from force-displacement curves that the AFM tip experiences repulsive forces at a distance of approximately 10-70 nm from the graphene surface. The authors also present SEM images of graphene suspended over holes in a SiN substrate onto which gold has been deposited. The low nucleation density of gold on the suspended graphene regions, compared to the regions of graphene supported by SiN, is attributed to vdW-repulsion. Since repulsive vdW-forces are mostly found in fluidic systems, this topic noteworthy and an interesting area of fundamental research that could have broad interest. That being said, it is very unclear whether the experimental evidence presented supports the conclusions and additional evidence is needed. Therefore, this work is not suitable for publication at the current stage.

The most important points are:

1. The low nucleation density of gold on suspended graphene is used as evidence for repulsive vdW forces. However, several factors can affect and explain the nucleation density on graphene, such as surface contamination (<https://doi.org/10.1002/sml.201101169>), roughness (<https://doi.org/10.1021/jp305521g>), and strain (<https://doi.org/10.1103/PhysRevB.52.R14380>). How can the authors rule out these factors?
2. Figure S6 shows that the vdW repulsion is expected to decrease with increasing graphene thickness. Hence, one would expect increasing gold nucleation density with increasing graphene thickness. However, this is the opposite trend as shown experimentally in Ref. 27 where the thickness dependent gold nucleation density on suspended graphene has been measured. The same trend is observed for gold on SiO₂-supported graphene: <https://doi.org/10.1021/ja909228n>, <https://doi.org/10.1021/nl9026605>. This shows that factors other than vdW repulsion determine the gold nucleation density on suspended graphene.
3. The manuscript lacks details regarding the AFM measurements. In AFM measurements in tapping mode, the tip can be driven into an attractive or repulsive regime by choice of the free amplitude: <https://doi.org/10.1103/PhysRevB.60.4961>. Therefore, the scanning parameters used in the measurements should be mentioned. That is both in order to enable reproduction of the experiment and to make sure that the authors considered how the details of the scanning parameters could affect the results.

Additionally, the following points should be considered:

4. The section on AFM measurements could benefit from further clarifications. On pg. 6 it says "Indeed, during the retraction process from a freestanding graphene surface, the tip remains to adhere to graphene at a large tip displacement, revealing that both graphene and AFM cantilever were bent before breaking the physical contact." How can a topography map be obtained if the tip and graphene interact so strongly?
5. On pg. 6 the authors write: "Nevertheless, there remains an atomically flat and contamination-free area of greater than 5 μm^2 near the center of suspended graphene". Annealing suspended graphene in Ar/H₂ has been shown to not be effective in cleaning the surface, see e.g., <https://doi.org/10.1021/nl203733r>, DOI: 10.1109/TNANO.2014.2365208. Furthermore,

thermal annealing graphene in high vacuum but then exposing the sample to ambient conditions leads to the redeposition of contamination: <https://doi.org/10.1116/1.5003034>. Have the authors conducted additional measurements to verify the graphene cleanliness or performed additional cleaning measures that goes beyond annealing in Ar/H₂? The graphene is likely contaminated with polymer residues from the transfer process and adsorbates from the air, therefore, any potential effects of this contamination in the measurements should be commented on.

7. On pg. 12 the authors write: "We deposited a small amount of gold (≈ 1.93 ng/mm²) at a rate of 0.01 nm/s on the sample surface at room temperature." For facilitating comparison with other studies the authors could state the nominal thickness of gold deposited.

Reviewer #2

(Remarks to the Author)

In this manuscript, the authors claim to have measured a large repulsive Casimir-van der Waals force when a suspended graphene sheet is used in conjunction with an AFM tip. The suspended graphene is treated as an optically anisotropic material and an alternative version of Lifshitz's equation is used to describe the interaction. If true, the result would be very significant. Unfortunately, there appear to be several contradictions between the results shown here and what has been found by other groups and reported in the literature. Below, we describe some of these issues, broken down between theory and experiment.

Theory:

o In the abstract, authors state, "very little is known about the many-body effects on their van der Waals (vdW) interactions" and "our theoretical framework taking into account the many-body Lifshitz formalism." These statements are confusing, because Lifshitz formalism does properly take into account many-body interactions.

o Theory has already been established for the Lifshitz force between Au and free-standing-graphene systems for separations >10nm, we for example: G. L. Klimchitskaya, V. M. Mostepanenko, and B. E. Sernelius, Two Approaches for Describing the Casimir Interaction in Graphene: Density-Density Correlation Function versus Polarization Tensor, Phys. Rev. B 89, 125407 (2014).

■ This reference predicts attraction, rather than repulsion. The authors must address this point.

■ The authors even cite a paper by Klimchitskaya, Mostepanenko, and Mohideen (Ref. 29), which also studies a similar system and finds repulsion, yet the differing models are not addressed.

o The authors present a new model for describing the Casimir interaction involving graphene. How does the modeling in the current manuscript compare to models used in the literature (not by the members of the groups on this paper) regarding the Casimir interaction involving graphene? See previous comment.

o Model for permittivity of graphene+vacuum layer appears to be derived for a multilayer stack in Ref. 28, not for a single layer. So is it then valid to use for a single layer? Why? It seems this model predicts a separation-dependent reflectivity on a graphene monolayer which appears contradictory and unphysical. Please discuss.

o Over what regimes (separation, photon frequency, incidence angle, ...) is their permittivity modeling valid?

o Page 4: claim of repulsion at all separations is contradictory, particularly at separations >300nm, with the abovementioned PRB paper. This should be addressed.

Experiments

o Fig. 2c: why is repulsion >10pN only present on 2 regions? Authors claim, "This is unsurprising because a large degree of interaction is internally absorbed by the soft nature of corrugated graphene," but I do not see why this necessarily results in a smaller repulsive force. There are other regions of the graphene that have a similar height profile (tall spots), but do not show repulsion >10pN

o Page 6: authors claim that the attractive feature at closest approach (which contradicts their theory) is due to capillary forces of a water layer. I think this is a somewhat simple hypothesis to test: collect force measurements with and without desiccant (that is change the humidity) in the system, and see if the separation at which the jump to contact occurs changes.

■ It seems critical to verify this hypothesis that suggests that the attractive feature seen at shortest separation, which is in direct contradiction with their theory, is caused by a water layer. Without testing this hypothesis, I wonder if the experiment is controlled enough and measurements are understood well enough in order to claim the seemingly repulsive feature is in fact repulsion and not some artifact. Theory by others contradict the experiments, and their own theory doesn't correctly describe it.

o Page 10: Why would a supposed increasing interaction area cause a linear scaling between deflection and displacement of sample towards tip? Could it be that it looks linear because the domain is sufficiently small?

o Seems like a stretch to say the experimental data can be quantitatively compared with theory given all of the above comments. Qualitatively they disagree at short separations.

o Figure S3: is there debris on the end of the probe? If so, and given it is of comparable size to the probe, that affect the measurement. In fact, the irregularity of the surface could easily explain the variation of force values (including sign).

o Section S2 discussion: It would be interesting to see the raw data. Optical interference in the AFM can cause an artifact to appear that mimics a repulsive peak. Further, the authors do a subtraction for the first harmonic, but make no mention of a second or third harmonic. These are sometimes small corrections, but they are claiming measurement of small forces. All of these issues need to be carefully considered.

o Figure S5b: Why are there multiple green and orange datasets?

Typos

o Page 5: 'formulism'  'formalism'

o Fig. 3e: legend should say 'BB' not 'BDP'

Reviewer #3

(Remarks to the Author)

The paper "Strong Repulsive Lifshitz-van der Waals Forces on Suspended Graphene" delves into the nature of surface forces in suspended graphene and their broader implications. It specifically investigates the repulsive Lifshitz-van der Waals (vdW) forces observed in suspended graphene, measured using atomic force microscopy (AFM). The study provides both theoretical and experimental insights into these forces, their magnitudes, and potential applications. Accordingly, the paper presents the direct measurement of significantly strong Lifshitz-vdW repulsive forces on suspended graphene. These forces are much stronger than previously reported Casimir-Lifshitz repulsions in fluid-based systems. However, for the paper to be considered for publication, several clarifications and improvements are necessary.

Exclusion of Other Forces:

The careful exclusion of other forces at the interface is needed. For example, electrostatic forces between the AFM tip and the samples are often seen in not ideally grounded systems. Experimental proofs should be provided.

Deformation Effects:

As mentioned in the paper, suspended graphene would undergo deformation when the AFM tip approaches the surface. Even within the suspending holes, the repulsive forces vary significantly. The reasons for this variation should be carefully examined and explained with evidence, not conjecture.

Surface Contaminations:

Surface contaminations generated from wet-transferred graphene are very likely to occur. The impacts on the measurements should be analyzed.

Impact of Surface Corrugation:

The paper mentions the presence of surface corrugation near the edges of suspended graphene but does not thoroughly analyze its impact on the measured forces.

Error Bars:

The error bars in figures 4a and 4b are extremely large. The accuracy and representativeness of the data should be explained.

Wetting Tendency:

For the wetting tendency difference of suspended graphene and graphene/SiNx, the strain effect and the surface roughness should be considered. The corrugations/wrinkles on graphene/SiNx obviously introduce a higher degree of nucleation sites, as seen in figure 5a, where nucleation forms lines. The surface roughness of SiNx would also introduce nucleation sites. The wettability due to surface roughness should be excluded.

Modeling Other 2D Systems:

If other 2D systems could be modeled and behave very differently, experimental measurements should be provided to examine the accuracy of the calculations.

Version 1:

Reviewer comments:

Reviewer #1

(Remarks to the Author)

I appreciate the authors' efforts and additional data that has been presented. However, some additions raise new questions which should be addressed before publication:

1. Pg. 8: "The last notable measured repulsive force were detected...". What is meant by the last "notable" measured repulsive force? Do the authors simply mean the last repulsive force before the tip experiences an attractive force? Also, should "were" actually be "was"?
2. On pg. 15 the authors write: "As a result, any electroneutral object approaching the surface could experience a repulsive force, smaller or larger, depending on the interacting configuration." Can they expand on what is meant by "interacting configuration" and which factors will make the force smaller or larger? Is this related to the results in Fig. 1(c, e) which has areas of the suspended graphene that does not display a repulsive force?
3. Fig. 5: The authors state in the rebuttal letter that their new result shown in Fig. 5 is not consistent with Ref. 30. Also, the thickness trend also goes against what is seen for SiO₂-supported Gr (references for this were given in the first reviewer assessment) as well as seen for other 2D materials such as MoS₂ on SiO₂. Although these latter samples are not suspended and therefore might not relate exactly to the samples studied here, it is nevertheless interesting that an opposite trend is observed. To allow readers to appreciate the link between this and other work, this should be mentioned and

discussed in the main text. The trend is not discussed in the updated manuscript, so the authors could also state that it matches their theoretical prediction, in support of the hypothesis.

4. Fig. 5: How was the layer thickness in each area assigned? Please include optical microscopy images of the sample and any other data in support of the determination.

5. Fig. 5: If the sample shown in Fig. 5 contains more suspended areas it would be good to show those as well (in the SI) to have an idea about differences in nucleation density across a sample. Comparing Fig. 5a with the original Fig. 5a there is also quite a large difference in nucleation density and particle size on monolayer graphene. The original image could also be shown to give a fuller picture.

6. Pg. 16: "The extremely low surface wettability of suspended graphene observed here can neither result from the atomic smoothness of graphene nor from any ultra high vacuum treatment, because the nucleation density of gold on single crystalline graphite cleaved in air remains high [54]". This sentence could be revised to improve clarity. Is the argument that it cannot be atomic smoothness because of the new result on Fig. 5? If so, the authors could state this. For the second part of the sentence, is the argument that the samples have been cleaved in air where a large nucleation density is expected, and nevertheless a low nucleation density is observed?

7. Conclusion: "effect" should be "effective".

Reviewer #2

(Remarks to the Author)

I commend the authors for the extra efforts in addressing the reviews; however, I do not believe that they have satisfactorily (and convincingly) shown that the observed phenomenon is related to the van der Waals force and not one of the many other effects and artifacts that may be present. As one simple example, when asked about electrostatic artifacts, the authors respond, "As the graphene was in close contact with gold, which was electrically grounded with the AFM tip throughout the measurement, one can ensure the exclusion of electrostatic interactions." Simply grounding the sample and tip does not eliminate electrostatic forces when dealing with real surfaces and interfaces, and there is a vast body of literature (both in surface science and within experimental Casimir measurements) that shows such connections do not necessarily remove electrostatic artifacts. Unfortunately, I cannot support publication of this manuscript.

Reviewer #3

(Remarks to the Author)

My concerns have been fully addressed. The manuscript has been substantially improved upon the revision. Thank you.

Reviewer #4

(Remarks to the Author)

I've been asked to share my thoughts regarding the disagreement between the authors and Reviewer #2, particularly the latter's concern that the authors have not convincingly shown that the observed forces originate from van der Waals interactions rather than other possible artifacts. The reviewer highlights the electrostatics issue, quoting the authors' statement that grounding both the AFM tip and the sample excludes electrostatic interactions — and I have to say, I think the reviewer raises a valid point.

In practice, simply grounding the tip and sample does not eliminate electrostatic forces. This is well documented in surface science and Casimir force literature. You can still have:

Patch potentials due to work function variation or local adsorbates;

Capacitive coupling based on tip geometry and proximity;

Surface contamination, even with careful transfer and annealing protocols;

And non-uniform charge distributions, which create localized electric fields that aren't canceled out by image charges.

So I agree that the electrostatic contribution hasn't been rigorously excluded, and more caution is needed here. That said, I do think the authors have made a genuine and thoughtful effort to address this. They redesigned the experiment to use grounded Au-coated membranes, tested multiple tip radii and materials, and consistently found similar repulsive force behavior. They also introduced a novel modeling approach, treating the suspended graphene as an effective birefringent medium, and their measurements appear reproducible with cleaner, smaller tips.

In the revised version, the authors do acknowledge that even if some electrostatic effect were present, it would be expected to result in attractive forces—not the repulsion they observe. While this doesn't rule out all artifacts, it does offer a plausible argument that electrostatics alone can't explain their data.

That being said, there are a few issues that I think one could have addressed:

Force sensitivity: The authors never quantify their detection limit, which is crucial. In air, pN-level forces are already approaching the limits of what a well-calibrated AFM can resolve. Their reported signals are very close to that floor, so they need to show — statistically — that what they're measuring is indeed above noise.

Tip radius: While SEM imaging helps, tip radius estimation is not an absolute value — it depends on imaging assumptions and can change due to wear or contamination.

Prior literature: It would be helpful if the authors acknowledged previous work in the attractive regime, *Langmuir* 2018, 34, 40, 11980–11988, which presents a careful study of forces on single-layer graphene and does a clearer job demonstrating monolayer identification.

In summary, I believe the authors are pointing to a potentially interesting effect, and they've clearly invested substantial effort in the revised work. But the burden of proof is high when reporting something this subtle and close to the measurement floor. More clarity on the detection threshold and a more rigorous treatment of possible electrostatic and topographic artifacts would make this a stronger, more convincing contribution.

Version 2:

Reviewer comments:

Reviewer #1

(Remarks to the Author)

I thank the authors for the thorough revisions. All my concerns have been addressed and I support publication.

Reviewer #4

(Remarks to the Author)

I commend the authors for presenting a bold and intriguing study. The effort to probe Lifshitz–van der Waals repulsion in suspended graphene using AFM and nucleation experiments is technically impressive and conceptually exciting. The revised manuscript shows a clear improvement in data presentation, and I appreciate the authors' extensive new experiments, including larger-area graphene, statistical validation, and thorough optical/Raman characterization.

That said, a few areas still merit attention:

The manuscript would benefit from a careful language revision. There are several grammatical issues and awkward phrasings throughout that occasionally obscure the meaning of key technical points. ex the word discussion on page 4

Electrostatics: As noted by Reviewer 2, grounding alone does not eliminate all possible electrostatic contributions, especially given known surface inhomogeneities and patch potentials. The authors have responded thoughtfully but could better acknowledge this limitation in the main text to resolve the issue with Reviewer 2.

Personally I have not observed this kind of repulsive behavior in my own AFM measurements, though I recognize differences in setup and conditions may explain this. If the authors are open to it, I would be pleased to independently attempt to reproduce these results using my system, as a cross-check could provide additional confidence in the robustness of the reported phenomenon.

Response Letter to the Reviewers

Response to Reviewer # 1

This work presents theoretical calculations predicting vdW-repulsion in a gold/graphene/vacuum system. To support this finding, the authors perform AFM measurements of suspended graphene using a gold-coated AFM tip and determine from force-displacement curves that the AFM tip experiences repulsive forces at a distance of approximately 10-70 nm from the graphene surface. The authors also present SEM images of graphene suspended over holes in a SiN substrate onto which gold has been deposited. The low nucleation density of gold on the suspended graphene regions, compared to the regions of graphene supported by SiN, is attributed to vdW-repulsion. Since repulsive vdW-forces are mostly found in fluidic systems, this topic noteworthy and an interesting area of fundamental research that could have broad interest. That being said, it is very unclear whether the experimental evidence presented supports the conclusions and additional evidence is needed. Therefore, this work is not suitable for publication at the current stage.

We thank the Reviewer for the constructive feedback. In order to properly address all questions raised by the Reviewers, we have carried out new experiments and rearrange the main text, in which we mainly focus on the experimentally observed repulsive force-displacement responses upon indentation, emphasizing on the connection and influence of the repulsion on flat suspended monolayer graphene. The theoretical analysis and the nucleation experiments are supportive to the main findings.

Please find our point-to-point response as follows.

The most important points are:

1. The low nucleation density of gold on suspended graphene is used as evidence for repulsive vdW forces. However, several factors can affect and explain the nucleation density on graphene, such as surface contamination (<https://doi.org/10.1002/sml.201101169>), roughness (<https://doi.org/10.1021/jp305521g>), and strain (<https://doi.org/10.1103/PhysRevB.52.R14380>). How can the authors rule out these factors?

We thank the Reviewer for raising this question. In order to elucidate the effects of surface contamination, roughness, and strain, we have carried out a new experiment by transferring a piece of exfoliated graphene containing monolayer, bilayer, and 4-layer regions on a holey SiNx membrane, followed by slow evaporation of gold on the sample. Before the deposition, the sample was annealed in hydrogen to remove the surface contamination. Although it is minimum, we could basically assume all areas have the same degree of surface contamination. In terms of surface roughness and strain, the bilayer and 4-layer graphene are even smaller than the monolayer counterpart, because the mechanical modulus increases with layer number.

Figure R1 (new **Fig. 5** in main text) compares the surface nucleation density on (a) monolayer, (b) bilayer, and (c) 4-layer areas, which suggest a clear trend: the nucleation density increases with the layer number. We can therefore infer that the surface contamination, roughness, and strain are not the dominant factors resulting in the substantially lowered wettability of monolayer graphene.

Note that in the Supplementary information, we have calculated the Lifshitz-vdW repulsion generated between gold and suspended graphene with layer number from 1 to 9 (Supplementary **Fig. S7**; see attached **Fig. R2**). The Lifshitz-vdW repulsion decreases with increasing the layer number, which can nicely explain the experimental findings presented in **Fig. R1**. Specifically, the reduction of Lifshitz-vdW repulsion could make the surface less “repulsive, thereby increasing the nucleation density.

Figure R1. SEM images comparing the nucleation density of gold on suspended (a) monolayer, (b) bilayer, and (c) 4-layer graphene deposited in an e-beam evaporator, at a constant rate of 0.01 nm/s and a nominal thickness of 0.1 nm. Scale bar: 1 μm .

Figure R2. Calculated vdW interaction energy for Vac/NL-Gr/Au system with various graphene layer numbers. With increasing graphene layer thickness, the repulsive response diminishes.

We have included the new results and discussion in the updated main text and supplementary information to elucidate the competing effects between Lifshitz-vdW repulsion, contamination, roughness, and strain.

Nevertheless, we understand that the new observation is not fully consistent with that presented in Ref. [1]. We think the deposition rate, or the kinetic energy of incident Au particles, also plays an important role. Clearly, more detailed and systematic experimentation is required to fully understand the phenomena, which is beyond the scope of this manuscript. As mentioned earlier, we would like to stress that the main focus of this manuscript is the generation of Lifshitz-vdW repulsion on suspended graphene upon AFM indentation, and the nucleation experiment of gold only represents a supporting evidence.

2. Figure S6 shows that the vdW repulsion is expected to decrease with increasing graphene thickness. Hence, one would expect increasing gold nucleation density with increasing graphene thickness. However, this is the opposite trend as shown experimentally in Ref. 27 where the thickness dependent gold nucleation density on suspended graphene has been measured. The same trend is observed for gold on SiO₂-supported graphene: <https://doi.org/10.1021/ja909228n>, <https://doi.org/10.1021/nl9026605>. This shows that factors other than vdW repulsion determine the gold nucleation density on suspended graphene.

We thank the Reviewer for pointing out the layer dependence. Following our earlier discussion, we have carried out a new experiment depositing gold on the same piece of exfoliated graphene containing monolayer, bilayer, and 4-layer regions. Our observation (**Fig. R1** or **Fig. 5** in main text) demonstrates an opposite trend to Ref. [27] (or Ref. [1] here), where the nucleation density increases with increasing the layer number. The new experiment actually support the vdW repulsion. We suggest more detailed and systematic experimentation is required to fully understand the phenomena, but it is beyond the scope of this manuscript.

For SiO₂-supported graphene, it is a different scenario, because the material stack A/m/B = SiO₂/Gr/Au does not obey the repulsion criteria $(\epsilon_A - \epsilon_m)(\epsilon_B - \epsilon_m) < 0$. The dielectric response of SiO₂ is greater than graphene. In other words, for samples of SiO₂-supported graphene, they are essentially attractive surfaces, and the nucleation behavior strongly depends on the surface properties, which have been thoroughly discussed in the literature suggested by the Reviewer. **The vdW repulsion does not exist in these SiO₂-supported graphene systems.**

Overall at least within the scope of this manuscript, the nucleation experiment on multilayer graphene carried out in our group does qualitatively support our claim of vdW repulsion.

3. The manuscript lacks details regarding the AFM measurements. In AFM measurements in tapping mode, the tip can be driven into an attractive or repulsive regime by choice of the free amplitude: <https://doi.org/10.1103/PhysRevB.60.4961>. Therefore, the scanning parameters used in the measurements should be mentioned. That is both in order to enable reproduction of the experiment and to make sure that the authors considered how the details of the scanning parameters could affect the results.

We would like to thank the Reviewer for his/her concern. As pointed out by the Reviewer, the choice of the free amplitude can drive the tip into an attractive or repulsive regimes in tapping mode. However, the PeakForce QNM mode established by Bruker, which is the mode applied throughout our AFM measurements, circumvents this exact problem, as stated on Page 8 in their Application Note #133 [2]. The detailed methodology for the Peakforce QNM mode please find in Ref. [3].

Specifically, the Peak QNM mode is an adaptive measurement procedure, adjusting itself within a set frame of parameters during the measurement. An analogous set of parameters to the free amplitude in the Peak QNM mode would be the ramp size, paired with the trigger threshold deflection. The ramp size determines the measurement range and the trigger threshold deflection determines the indentation position for the tip to be retracted, in our case set to 0.25 V resulting in maximal indentation forces up to 3.18 nN. During the indentation, other parameters are adjusted adaptively to guarantee consistent measurement conditions without yielding artifacts.

Please find the typical values of measurement parameters used in Peakforce QNM mode below.

Tip - Radius (nm)	Deflection Sensitivity (nm/V)	Spring Constant (N/m)	Trigger Thres. Deflection 0.25V (nN)	Ramp Size (nm)	Ramp Rate (Hz)
Au - 30	69.679	0.16136	2.81	771	1.02
SiNx - 20	75.567	0.11670	2.20	771	1.02
Au - 13	86.484	0.14710	3.18	900	0.88

Additionally, the following points should be considered:

4. The section on AFM measurements could benefit from further clarifications. On pg. 6 it says "Indeed, during the retraction process from a freestanding graphene surface, the tip remains to adhere to graphene at a large tip displacement, revealing that both graphene and AFM cantilever were bent before breaking the physical contact." How can a topography map be obtained if the tip and graphene interact so strongly?

We thank the Reviewer for the thoughtful comment. **Figure R3** on the next page illustrates the process for the extraction of topography and force-displacement responses. In short, we actually extracted both force and topography maps based on the $F - d$ responses **during the approach process**, before establishing the contact. Therefore, the strong adhesion experienced during the retraction process (after establishing the contact) does not influence the maps generation.

Specifically, **Figure R3a-c** presents a typical force-displacement response recorded during the tip approach and retraction process. **Fig. R3b** and **Fig. R3c** highlight the rescaled responses during the approach process, corresponding to the purple and red dashed boxes in **Fig. R3a** and **Fig. R3b**, respectively.

As shown in **Fig. 4d**, the surface force map is generated by reading the force value, F , experienced by the tip at a given displacement value d (in this case, $d = 10$ nm at Stage I), before establishing the contact (Stage II, or $d = 0$), during individual force-displacement measurement at a lateral position point (x, y) on the sample surface. Namely, $F|_{d=10nm}(x, y)$ correspond to the force maps shown in **Figs. 1c** and **1e**. As such, one can guarantee that the detected force value is experienced before making the contact, purely resulting from the interaction between the AFM tip and suspended graphene.

The topography map, on the other hand, is generated by directly monitoring the cantilever movement during the approach process. At a given a lateral position point (x, y) , we recorded the vertical position, z , when the cantilever is deflected, corresponding to the force minimum during the approach process (the contact point at Stage II), as illustrated in **Fig. R4d**. After adjusting to the zero point, the nominal vertical position as a function of lateral position, $\bar{z}(x, y)$, corresponds to the topography maps shown in **Figs. 1d** and **1f**.

Finally, we would like note that the adhesion force detected during the retraction process from the suspended graphene is not particularly strong. As mentioned in the main text, the forces required to break the contact (10 – 20 nN) are approximately identical for retracting from the surfaces of (i) suspended graphene and (ii) SiNx-supported graphene, as shown in Supplementary **Fig. S4** (also **Fig. S4** attached below). The substrate-independent nature of this force suggests that this force may be the capillary force, resulting from the condensation of water between tip and graphene surface, as discussed in the main text.

In summary, we would like to stress again that the force-displacement responses during the retraction process does not involve in the generation of force and topography maps.

Figure R3. Schematic diagram illustrating the generation of force and topography maps during the approach process of indentation. **(a)** Representative force-displacement curve including both approach (green curve) and retraction (red curve) responses. **(b)** The rescaled approach response corresponding to the purple dashed box in panel (a). **(c)** The rescaled approach response corresponding to the red dashed box in panel (b). **(d)** Schematic diagram illustrating Stages I and II, corresponding to the displacement values

$d = 10$ nm and $d = 0$, during the approach process before establishing the contact, which generates the force and topography maps, respectively.

Figure R4. Representative force-displacement responses experienced by the AFM tip during the approach (green curve) and retraction (red curve) process on (a) suspended graphene, and (b) SiNx-supported graphene. The insets magnify the force responses at small displacements.

5. On pg. 6 the authors write: "Nevertheless, there remains an atomically flat and contamination-free area of greater than $5 \mu\text{m}^2$ near the center of suspended graphene". Annealing suspended graphene in Ar/H₂ has been shown to not be effective in cleaning the surface, see e.g., <https://doi.org/10.1021/nl203733r>, DOI: 10.1109/TNANO.2014.2365208. Furthermore, thermal annealing graphene in high vacuum but then exposing the sample to ambient conditions leads to the redeposition of contamination: <https://doi.org/10.1116/1.5003034>. Have the authors conducted additional measurements to verify the graphene cleanliness or performed additional cleaning measures that goes beyond annealing in Ar/H₂? The graphene is likely contaminated with polymer residues from the transfer process and adsorbates from the air, therefore, any potential effects of this contamination in the measurements should be commented on.

We thank the Reviewer for raising the concern. We agree with the Reviewer that the Ar/H₂ annealing is indeed insufficient to remove the polymer contaminant for PMMA-based graphene transfer process, as suggested in the literature pointed out by the Reviewer. In fact, in our group, we had also extensively examined the PMMA-based transfer process and reached the same conclusion. This challenge had motivated us to explore alternative graphene transfer process that yields ultraclean surface.

Inspired by Ref. [4], we have used the cellulose acetate butyrate (CAB) as a sacrificial layer to transfer the exfoliated graphene. Our tests have suggested that it is significantly easier to remove CAB in acetone, leaving only very small amount of polymer residue after transfer, in line with another recent report [5]. As a result, the post process of Ar/H₂ annealing becomes very effective, yielding nearly contamination-free graphene samples. Right after annealing, the samples were kept in vacuum before subsequent experiments or characterization.

More specifically, Ar/H₂ annealing allows the removal of: (i) large specks of organic residue after acetone rinsing from both suspended and substrate-supported graphene (see **Figs. R5a** and **R5b**), and (ii) airborne contaminants that gradually absorb on the surface after long-term ambient exposure. Note that any SEM-based inspection should be also avoided before the AFM measurement, in order to prevent any electron beam-induced hydrocarbon deposition [6]. **Figure R5c** presents an SEM image for an area of suspended graphene after Ar/H₂ annealing, revealing that only very few polymer residues remained (highlighted in white circles), which are nearly negligible in AFM measurements.

On the other hand, we would like to point out that **the Lifshitz-vdW repulsion is a mesoscale force, insensitive to the surface property of graphene**. As discussed in our response to the first point raised by the Reviewer, the calculated repulsive force remain strong even for multilayer graphene (see **Fig. R2**). In other words, even there is a small degree of polymer residue or airborne contamination deposited on graphene surface, the vdW repulsion would remain to occur.

Figure R5. Ar/H₂ annealing of graphene transferred by using the CAB-based protocol. Optical micrographs for graphene transferred on to a holey SiN_x membrane (**a**) before and (**b**) after Ar/H₂ annealing, showing that the majority of polymer residues were removed on both suspended and substrate-supported graphene. (**c**) Representative SEM image for an area of suspended graphene after Ar/H₂ annealing, showing nearly contamination-free graphene surfaces. Scale bar: 1 μm.

In summary, in all samples considered in this report, we have utilized the CAB-based graphene transfer protocol, which maximally remove polymer residues, so that the Ar/H₂ annealing can yield nearly contamination-free graphene surfaces. Moreover, the Lifshitz-vdW repulsion is a mesoscale force, insensitive to a small amount (if there is any) of surface contaminants. Altogether, we conclude that the effects of organic residues are nearly negligible to the measurement of Lifshitz-vdW repulsion.

6. On pg. 12 the authors write: "We deposited a small amount of gold ($\approx 1.93 \text{ ng/mm}^2$) at a rate of 0.01 nm/s on the sample surface at room temperature." For facilitating comparison with other studies the authors could state the nominal thickness of gold deposited.

We thank the Reviewer for pointing this out. We have added the nominal thickness value of the gold in the manuscript, which is 0.1 nm, as we aimed to observe the nucleation behavior at a very early stage.

In the end, we would like to draw the Reviewer's attention that in response to other Reviewer's comments, we have carried out new measurements on another piece of suspended monolayer graphene transferred to a gold-coated holey SiN_x membrane, and the sample was electrically grounded during the indentation. A strong vdW repulsion was consistently detected in the flat area of the suspended graphene (see **Fig. R6**, corresponding to new **Fig. 1** in main text). We invite the Reviewer to look into our new results and analysis.

Again we very appreciate the Reviewer's constructive comments and feedback.

Figure R6. Direct measurement of the Lifshitz-vdW repulsion on suspended graphene. **(a)** Schematic diagram for the AFM measurement of the interaction forces experienced by a gold-coated AFM tip approaching graphene transferred to a holey membrane. **(b),(c)** Representative topographical **(b)** and corresponding surface force **(c)** maps for a 33 nm radius gold-coated AFM tip interacting with a sheet of micromechanically exfoliated graphene transferred on to a LPCVD-grown SiN_x holey membrane. The circle area of diameter of approximately 5 μm corresponds to suspended graphene. **(d),(e)** Representative topographical **(d)** and surface force **(e)** maps generated using a 13 nm radius gold-coated AFM tip that approached another piece of graphene transferred onto Au-coated SiN_x holey membrane.

Response to Reviewer # 2

In this manuscript, the authors claim to have measured a large repulsive Casimir-van der Waals force when a suspended graphene sheet is used in conjunction with an AFM tip. The suspended graphene is treated as an optically anisotropic material and an alternative version of Lifshitz's equation is used to describe the interaction. If true, the result would be very significant. Unfortunately, there appear to be several contradictions between the results shown here and what has been found by other groups and reported in the literature. Below, we describe some of these issues, broken down between theory and experiment.

We thank the Reviewer for the constructive feedback. In order to properly address all questions raised by the Reviewers, we have carried out new experiments and rearrange the main text, in which we mainly focus on the experimentally observed repulsive force-displacement responses upon indentation, emphasizing on the connection and influence of the repulsion on flat suspended monolayer graphene. The theoretical analysis and the nucleation experiments only play a supportive role to the main findings. In particular, we acknowledge that our theoretical framework has a certain degree of limitation. While a more accurate description of the system is possible, it is beyond the current scope of this manuscript. We would appreciate the Reviewer's understanding.

Please find our point-to-point response as follows.

Theory:

In the abstract, authors state, "very little is known about the many-body effects on their van der Waals (vdW) interactions" and "our theoretical framework taking into account the many-body Lifshitz formalism." These statements are confusing, because Lifshitz formalism does properly take into account many-body interactions.

We thank the Reviewer for pointing this out. We agree with the Reviewer that the statement was rather confusing. We have changed the statement as follows.

"Despite recent observations of wetting transparency and remote epitaxy in substrate-supported 2D monolayers, the intrinsic surface forces of suspended 2D materials remain vague, including the many-body effects of surrounding vacuum."

Theory has already been established for the Lifshitz force between Au and free-standing-graphene systems for separations > 10 nm, we for example: G. L. Klimchitskaya, V. M. Mostepanenko, and B. E. Sernelius, Two Approaches for Describing the Casimir Interaction in Graphene: Density-Density Correlation Function versus Polarization Tensor, Phys. Rev. B 89, 125407 (2014).

This reference predicts attraction, rather than repulsion. The authors must address this point.

We would like to thank the Reviewer for pointing out this important reference. It is indeed very insightful. In the new version of manuscript, we have significantly modified the manuscript, by not only acknowledging the reference pointed out by the Reviewer, but also addressing the main differences between the two approaches.

Specifically, we agree with the Reviewer that when treating graphene as a static interface of either infinitesimal [7] or finite size [8] would result in attraction. This approach is proper particularly for systems having graphene supported by a hard solid substrate, so that upon reducing the separation, the generated

force does not deform graphene. In other words, as long as the “effective volume” of graphene remains independent of separation, the static-interface approach makes perfect sense.

However, in the suspended graphene systems considered here, the indentation of AFM tips having radii comparable to the separation will certainly deform graphene, which is a sheet of one-atom thick membrane having remarkable flexibility, comparable to soft fluid. In a simplified one-dimensional Lifshitz formalism, the “effective volume of graphene” upon indentation becomes separation dependent. The configurations considered in the theories pointed out by the Reviewer may not capture every nuance of the suspended graphene.

Notably in the literature, there are certain specific geometries known to induce repulsion for materials. For example, Refs. [9,10] demonstrate the repulsive forces generated between anisotropic particles and a holey metallic plate. There are a number of computational studies predicting the generation of repulsive Casimir interactions in the interlocked geometries [11,12] and the sphere-plane geometries between two perfect conductors [13]. None of these systems involves flat interfaces.

The Lifshitz-van der Waals forces of the suspended graphene systems are particularly hard to model, considering both graphene’s dynamic and flexible nature, as well as the nonplanar geometry. We had been inspired by the experimental findings of high bending radii of up to 1 nm for few-layer graphene [14] and the theoretical treatment by Milton et al. [15], where a medium of high permittivity intervening a low-permittivity space yields repulsion. Specifically, in this model, flexible graphene and vacuum are treated as a single, unified medium, as illustrated in **Fig. R7 (Fig. 3a** in main text). The effective medium intervening graphene and surrounding vacuum upon the indentation of AFM tip behaves analogous to a fluid immersion system. We consider that the suspended graphene surface follows a similar notion being pulled into the space between itself and the AFM tip upon approach, as depicted in **Fig. 3b**.

In fact, we have gained more insight while comparing the graphene’s polarizability α_{Gr} computed by our expression with that suggested in the reference by the Reviewer. The implementation of different permittivity descriptions yield similar trend within the separation range of interest compared to ours. We have included both new insights and evaluations to the manuscript with proper recognition to other models of interaction. Please see **Fig. 4a** in the main text, which is attached on next page.

With the above rationale in mind, although our effective medium approach might a bit simplified at first glance, the main objective of this manuscript is to report the repulsion we measured on suspended graphene upon the indentation of an AFM tip. As mentioned earlier, the theoretical part represents a possible treatment explaining and supporting the observed vdW repulsion. We believe a more advanced theoretical framework could make better prediction but is beyond the scope of this manuscript.

Figure R7 (Fig. 3a in main text). Schematic illustration the origin of the assumption of treating graphene and the intervening space as an effective fluid immersion medium, where graphene is significantly distorted upon AFM tip indentation.

- The authors even cite a paper by Klimchitskaya, Mostepanenko, and Mohideen (Ref. 29), which also studies a similar system and finds repulsion, yet the differing models are not addressed.

We would like to point out although the system in Ref. [16,17] might look similar, it is inherently different as it dealt with graphene supported on SiO_2 . On one hand, following our discussion above, the model did not take into account the mechanical flexibility of graphene and the intervening nature of graphene and surrounding vacuum upon AFM indentation. On the other hand, SiO_2 has a high dielectric response at high frequencies, so that the system, $A/m/B = \text{SiO}_2/\text{Gr}/\text{Au}$, does not obey the repulsive inequality (1) in the main text: $(\epsilon_A - \epsilon_m)(\epsilon_B - \epsilon_m) < 0$. The dielectric stack system will always generate attractive force, regardless the model used. In fact, a similar behavior can be observed in **Fig. R8** (**Fig. 4a** in main text), where when $A/m/B = \text{SiN}_x/\text{Gr}/\text{Au}$, the vdW forces are always attractive at all separations.

Figure R8 (**Fig. 4a** in the updated main text) | Comparison of Lifshitz theory-calculated vdW forces per unit area as a function of separation z using the effective media approach for $A/m/B = \text{Vac}/\text{Gr}/\text{Au}$, $\text{Vac}/\text{Gr}/\text{SiN}_x$, $\text{SiN}_x/\text{Gr}/\text{Au}$, $\text{SiN}_x/\text{Gr}/\text{SiN}_x$ and $\text{SiO}_2/\text{BB}/\text{Au}$ and $\text{Vac}/\text{Gr}/\text{Au}$ using the density-density (den-den) correlation function for the description of the polarizability from is also attached.

- The authors present a new model for describing the Casimir interaction involving graphene. How does the modeling in the current manuscript compare to models used in the literature (not by the members of the groups on this paper) regarding the Casimir interaction involving graphene? See previous comment.

We would like to thank the Reviewer for their interest in our model used to inform our experimental data. Here we would like to refer to our earlier discussion regarding the difference of our suspended graphene system with those considered in the literature. In short, we have used the effective medium approach to describe the geometric complexity and intervening nature of flexible graphene. We would like to, however, draw the Reviewer’s attention that this model is used solely for trying to understand the experimental data presented in this manuscript. We are open to a better model describing the polarizability of graphene taking account of the flexibility of graphene.

Furthermore, the claim of suspended graphene being a repulsive surface is also indirectly supported by the experimental observation of graphene’s non-wettability upon the evaporation of gold. In addition to **Fig. 5** shown in our manuscript, other experimental groups also observed that neither gold nor other metals can adhere to the surface of suspended monolayer graphene [19,20].

- Model for permittivity of graphene+vacuum layer appears to be derived for a multilayer stack in Ref. 28, not for a single layer. So is it then valid to use for a single layer? Why? It seems this model predicts a separation-dependent reflectivity on a graphene monolayer which appears contradictory and unphysical. Please discuss.

The permittivity for the effective medium graphene+vacuum approach was indeed derived originally from a multilayer stack. This is due to the limitation in DFT calculation to have periodic boundary conditions. However as discussed in reference [18], the DFT calculations suggest that the dielectric function of a monolayer 2D material is an ill-defined property, strongly depending on the size of surrounding vacuum due to the long-range nature of dielectric screening. The calculations suggest that a vacuum size of considerably larger than 10 nm is required to obtain a size-independent dielectric constant. Considering the vdW separations considered in this report are of similar length scale (6 – 60 nm), it is actually not unphysical to have separation-dependent dielectric constant of graphene.

Accordingly, following the suggestion of Ref. [18], our dielectric responses of graphene were calculated from the separation-independent polarizability functions extracted from DFT calculations. The separation dependence (or the vacuum-size dependence) for the in- and out-of-plane dielectric constants are therefore unique, without the error resulting from periodic boundary conditions.

Furthermore, as discussed earlier, another important reason for us to use the effective graphene+vacuum approach is the fact of graphene’s ultraflexibility and the intervening nature with surrounding vacuum upon AFM tip indentation (**Fig. R7**).

In fact, the separation dependency is very similar to the description of the wave vector dependency k_{\perp} for modeling graphene permittivity found in Ref. [7]. One can quickly do a dimensional analysis for the expressions α^{\parallel} of graphene for both the density-density correlation function and polarization tensor approach. For frequencies $\xi_l > v_F k_{\perp}$ where v_F is the Fermi velocity, we can see that both the Eq. (18) for the density-density correlation and the first term of the 00 –component of the polarizability expression Π_{00} in Equation (6) scale with k_{\perp} , which can be interpreted as inverse scaling with the separation. To further illustrate this point, we have carried out calculations using the permittivity expressions of graphene from the density-density correlation function approach from Ref. [7]. Please see the blue (Vac/Gr/Au den-den) curve in **Fig. R9** (**Fig. 4b** in main text).

Figure R9 (**Fig. 4b** in the updated main text) | Comparison of Lifshitz theory-calculated vdW forces per unit area as a function of separation z using: (i) the distance-dependent dielectric responses calculated from

the DFT-calculated distance-independent polarizability functions (Vac/Gr/Au; purple curve) and (ii) the density-density (den-den) correlation function approach adopted from Ref. [7] (Vac/Gr/Au; blue curve).

The comparison in **Fig. R9** shows that both descriptions of permittivity yield repulsion for the separation range of interest considered in this work, $6 < d < 60$ nm. The usage of an alternative description for the permittivity only results in a shift of the potential by a few nm in separation. Both models capture the trend. In other words, the vdW repulsion is generated as long as graphene is treated as an intermediate media, irrespective of the permittivity model. Note that although we were not able to computationally implement the polarization tensor approach, we can still make a reasonable estimation based on the information given **Fig. 4** from Ref. [7]. Indeed, the density-density correlation function tends to overestimate the interaction compared to the polarizability tensor approach from 8.5% at 10 nm by up to 41.8 % at separations larger than 400 nm, suggesting that the polarizability tensor approach should be close to our model.

An interesting observation for the calculated force using the density-density correlation is the sudden transition from repulsion to attraction at short separations < 3 nm. The attraction seems to be caused by an increase in polarizability at small separation due to a shift in relation between wavevector and frequency for the Fermi velocity. At this point we are not able to assert if it is physical, but it seems support our observation at short separations.

- *Over what regimes (separation, photon frequency, incidence angle, ...) is their permittivity modeling valid?*

We thank the Reviewer for the question. As the permittivity responses come from DFT-calculated polarizability of graphene, which assume the chemical interactions are negligible, the valid separation should be at least larger than 1 nm. The frequency range spans over from the 0th Matsubara frequency up to the 1001st one at 300 K, up to a frequency of $2.468 \times 10^{16} \frac{\text{rad}}{\text{s}}$. As for the incidence angles, our calculations only consider the orthogonal wave vectors for the interaction.

- *Page 4: claim of repulsion at all separations is contradictory, particularly at separations > 300 nm, with the above mentioned PRB paper. This should be addressed.*

We thank the Reviewer from pointing this out. Indeed we did not detect any repulsion at a separation of greater than 100 nm. We have removed the statement of repulsion at all separations and specifically mentioned that the range of interest is $6 < d < 60$ nm. All the calculations dealt with the interactions for $d < 100$ nm (see **Fig. R8**).

Experiments:

- *Fig. 2c: why is repulsion > 10 pN only present on 2 regions? Authors claim, "This is unsurprising because a large degree of interaction is internally absorbed by the soft nature of corrugated graphene," but I do not see why this necessarily results in a smaller repulsive force. There are other regions of the graphene that have a similar height profile (tall spots), but do not show repulsion > 10 pN*

We thank the Reviewer for the question. Here we would like to elaborate how the surface properties of graphene, including its highly flexible nature, influence the repulsion detected by the AFM tip. As

mentioned in the manuscript, the repulsion was consistently measured on flat suspended graphene only. In order to further exam the flatness requirement, we have carried out a new set of AFM measurements, as shown in **Fig. R10**. In addition, to exclude any scenario of electrostatic interactions, we have transferred graphene onto a gold-coated holey SiNx membrane, which was electrically grounded throughout the AFM indentation and scanning. In this sample, we were able to transfer graphene free of corrugation as one can confirm from the topographical map found in **Fig. R10a**. The suspended area resembles a parabolic surface, with relatively flat region exactly located within the center of the pore. Combining with the surface force map shown (**Fig. R10b**), we can clearly observe that the repulsive forces occurred at the flat region of the suspended graphene. To further quantify the flatness required to exhibit repulsion, we have analyzed the surface profile of the suspended region, with respect to the surface force. **Figures R10 c** and **d** superimpose the repulsive force values measured over two representative cross-sectional cuts of the suspended area.

Figure R10 (Fig. 1d-g in the updated main text) | Generation of repulsive force on flat area of suspended graphene. Topographical (a) and surface force (b) maps generated using a 13 nm radius gold-coated AFM tip that approached another piece of graphene transferred onto Au-coated SiNx holey membrane. The surface force maps present the vertical force values experienced by the AFM tip at a displacement d of approximately 10 nm before establishing the contact. The areas enclosed by white dashed lines correspond to the regions experiencing repulsive forces of ≥ 10 pN. (c), (d) Representative cross-sectional cuts through the respective repulsive domains in (a) and (b) combining the topographic and surface force profiles. A parabolic function $f_p(x)$ was used to fit the topographic profile, suggesting that the repulsion emerges when graphene is relatively flat, with the maximum surface gradient, $\left\| \max \left(\frac{df_p}{dx} \right) \right\| \leq 0.01$.

To quantitatively examine the effects of surface flatness, we have fitted local surface profile with a parabolic function, $f_P(x) = a(x - b)^2 + c$, at a given position x . Together with the map of measured repulsive forces, we have clearly identified that the repulsion emerges when graphene is relatively flat, with the maximum surface gradient, $\left\| \max\left(\frac{df_P}{dx}\right) \right\| \leq 0.01$. One can also clearly observe how the interacting area amplifies the repulsive feedback. Note that the overall response in the new measurements were weaker, because the new gold-coated AFM tip has a smaller radius (13 nm), thus resulting in a smaller interacting area.

The height profile in itself is irrelevant for the success of the measurement, as one can see from **Fig. R11**. **Figure R11a** presents the repulsive force responses along the vertical cross-sectional cut through the smaller repulsive area found in **Fig. 1b-c**. For comparison purposes, **Fig. R11b** shows the responses along another vertical cross-sectional cut through a non-repulsive region. Although both are of similar height in certain regions, they do not exhibit similar repulsive feedback. Indeed, in **Fig. R11a**, the local ripples having surface corrugation sometimes form convex domains with sufficiently flat area allowing the detection of repulsive forces. As long as there are no sufficiently large flat domains, like **Fig. R11b**, the repulsive responses could not be detected. Together with **Fig. R10**, one can conclude that the indentation of an AFM tip of finite tip radius can only resolve the orthogonal forces, which demand a relatively flat surface. When the surface has domains of a degree of inclinations, the indentation of an AFM tip would most likely result in a lateral displacement for either the AFM tip or the graphene surface.

We hope the quantitative analysis presented in **Figs. R10** and **R11** clearly address the Reviewer’s question.

Figure R11 | Detection of repulsive forces on corrugated graphene. Representative cross-sectional cuts through the corrugated graphene sample shown in **Fig. 1b,c**, including (a) a cut passing through a repulsive area, and (b) a cut passing through a corrugated area without the detection of repulsion.

Page 6: authors claim that the attractive feature at closest approach (which contradicts their theory) is due to capillary forces of a water layer. I think this is a somewhat simple hypothesis to test: collect force measurements with and without desiccant (that is change the humidity) in the system, and see if the separation at which the jump to contact occurs changes.

It seems critical to verify this hypothesis that suggests that the attractive feature seen at shortest separation, which is in direct contradiction with their theory, is caused by a water layer. Without testing this hypothesis, I wonder if the experiment is controlled enough and measurements are understood well enough in order to claim the seemingly repulsive feature is in fact repulsion and not some artifact. Theory by others contradict the experiments, and their own theory does not correctly describe it.

We thank the Reviewer for the helpful suggestions. We indeed try to reduce the relative humidity of the chamber during the new run of measurements for the purpose of this review by using not only desiccants but also by flushing the chamber with nitrogen. The flushing was done prior to start the measurement in order not to disrupt the delicate mechanics of the AFM. Unfortunately we were only able to reduce the relative humidity from 45% to approximately 20%, due to a large volume of AFM enclosure. On the other hand, our measurement took several hours, which leads to an increase in humidity over time.

On the other hand, as we discussed in **Fig. R9**, the calculated force using the density-density correlation also shows a sudden transition from repulsion to attraction at short separations < 3 nm. The attraction seems to be caused by an increase in polarizability at small separation due to a shift in relation between wavevector and frequency for the Fermi velocity. At this point we are not able to assert if it is physical, but it seems support our observation at short separations. Again we would like to focus on the repulsive forces generated at intermediate separations, and the theoretical model represents a first supportive trial to explain our experimental findings. We believe further implementation of more accurate model describing the permittivity of 2D monolayer could improve the model at short separations.

Page 10: Why would a supposed increasing interaction area cause a linear scaling between deflection and displacement of sample towards tip? Could it be that it looks linear because the domain is sufficiently small?

The linear scaling of the deflection, or the experienced repulsive forces, with respect to the displacement results from the increase of interacting area upon the indentation of the AFM tip. As we discussed earlier, owing to its atomic thickness, the mechanical flexibility of graphene makes it easily deform when experiencing a repulsive force. Together with the fact that the hemispherical tip radius (10 – 30 nm) is comparable to the separations of interest (6 – 60 nm), the graphene deformation is clearly playing a role. In other words, upon the approach of an AFM tip, we assume: (i) the interacting area increases, scaling with the height of the hemisphere immersed in the surface, and (ii) the separation between the tip and graphene remains nearly constant. As such, the force is approximately linearly scaled with the tip displacement. We have included a short animation showing this dynamic process. The observation that the width of this linear region is identical to the tip radius nicely support the scenario proposed here.

In order to further support the physical picture, we have carried out another set of measurements using a gold-coated AFM tip of smaller tip radius (approximately 13 nm, see **Fig. R12a**). Note that in the new set of measurements, we transferred graphene on top of gold-coated SiNx holey membrane, so that both graphene and AFM tip are electrically grounded throughout the measurement to exclude the electrostatic effects. In addition to the consistent observation of repulsive response (**Fig. R12b**), we have observed that the width of the linear region is also reduced to ~ 13 nm (**Fig. R12c**), supporting our assumptions.

Figure R12 | A new set of force measurements using a gold-coated AFM tip of smaller radius. (a) Tilted SEM image of the AFM tip after the new set of force measurements. The tip remained clean and has a tip radius of approximately 13 nm (Scale bar: 100 nm). **(b)** 2D histogram of 144 force-displacement measurements taken from the center of the new sample. A repulsive force of 7.5 ± 5.7 pN was measured at an average separation of approximately 7.5 nm. The reduction of repulsive force results from a smaller interacting area. **(c)** The magnified force-displacement response showing a clear linear region of width of ~ 13 nm, consistent with the tip radius.

- Seems like a stretch to say the experimental data can be quantitatively compared with theory given all of the above comments. Qualitatively they disagree at short separations.

We agree with the Reviewer that our model cannot describe the responses at very small separations. In the updated version of main text, we have clearly pointed out the limitations of our model and its supporting

role for understanding the force-displacement responses experienced upon indentation on suspended graphene for $6 < d < 60$ nm, which is the main focus of this manuscript. Finding a model that can quantitatively describe the force-displacement responses at all separations is beyond the scope of this manuscript.

As discussed earlier, we believe that a more proper description for graphene's permittivity at such small separations would require the wavevector dependent polarizability presented in Ref. [7]. As shown in **Fig. R9**, the calculated force using the density-density correlation seems to catch the trend of sudden transition from repulsion to attraction at short separations < 3 nm. At this point we are not able to assert if it is physical, but it seems support our observation at short separations.

Figure S3: is there debris on the end of the probe? If so, and given it is of comparable size to the probe, that affect the measurement. In fact, the irregularity of the surface could easily explain the variation of force values (including sign).

We thank the Reviewer for raising this question. As discussed earlier in **Fig. R12a**, have carried out a new set of experiments using a gold-coated AFM tip of smaller tip radius, which consistently showed repulsion. We intentionally examined the tip after all measurements using SEM, showing that the tip remained clean.

More specifically, in the new measurements, there are three main findings as follows. (i) The width of linear region is consistent with the tip radius, endorsing our picture of graphene deformation upon indentation (see our earlier discussion for details). (ii) The AFM tip remains clean without debris after all measurements (**Fig. R12 a**). (iii) As shown in **Figs. R12 b** and **c**, the evaluation of 144 force-displacement responses at the center of the suspended graphene reveals a repulsive force of 7.5 ± 5.7 pN at an average separation of approximately 7.5 nm before making contact. These values allow us to estimate the geometric parameters, the effective radius R_{eff} and area of interaction A_{max} of 18nm and 4254 nm² respectively. Considering the smaller size of tip, the values are surprising intact with the physical picture presented in **Fig. 4e**.

In summary, given the observations that both sets of experiments yielded consistent results, we can safely assume that the debris did indeed not affect the initial set of measurements.

Section S2 discussion: It would be interesting to see the raw data. Optical interference in the AFM can cause an artifact to appear that mimics a repulsive peak. Further, the authors do a subtraction for the first harmonic, but make no mention of a second or third harmonic. These are sometimes small corrections, but they are claiming measurement of small forces. All of these issues need to be carefully considered.

We thank the Reviewer for pointing this out. Following our earlier response, we have attached 16 individual raw force-displacement measurements without background subtraction in **Fig. R13** on the next page. Although we agree with the Reviewer that the higher order harmonics can indeed influence the measurement, we believe that this is not the case here. The influence caused by the optical interference within our area of interest is substantially diminished as all of them are usually located within the center of the pore, in which almost no light is back reflected. This can be seen from both the amplitude of the first harmonic being much smaller than the maximum repulsive feedback in **Fig. R13** extracted from both sets of experiments.

Figure R13 | Original force-displacement responses without optical interference correction. The representative original force-displacement responses are chosen from the 144 measurements used for the generation of 2D histograms shown in (a) **Fig. 2a** in the main text and (b) **Fig. R12b** (see Page 17).

Figure S5b: Why are there multiple green and orange datasets?

We thank the Reviewer for pointing this out. The extra green and orange points were artifacts generated during figure rendering. We have corrected it in the new SI file (**Fig. S8b**) and attached below.

Figure R14 | Newly rendered **Fig. S8b** in the Supplementary Information after removing the artifacts.

Typos:

- Page 5: 'formulism' → 'formalism'
- Fig. 3e: legend should say 'BB' not 'BDP'
- Page 10: 'plat' → 'flat'

We would like to thank the Reviewer again for going through our manuscript carefully and pointing out finding these typos. We have corrected all of them in the new main text. We hope the Reviewer could recognize our effort made during the revision stage of this manuscript.

Response to Reviewer # 3

(Remarks to the Author): The paper "Strong Repulsive Lifshitz-van der Waals Forces on Suspended Graphene" delves into the nature of surface forces in suspended graphene and their broader implications. It specifically investigates the repulsive Lifshitz-van der Waals (vdW) forces observed in suspended graphene, measured using atomic force microscopy (AFM). The study provides both theoretical and experimental insights into these forces, their magnitudes, and potential applications. Accordingly, the paper presents the direct measurement of significantly strong Lifshitz-vdW repulsive forces on suspended graphene. These forces are much stronger than previously reported Casimir-Lifshitz repulsion in fluid-based systems. However, for the paper to be considered for publication, several clarifications and improvements are necessary.

We thank the Reviewer for her/his constructive comments. Please find our response as follows.

1. Exclusion of Other Forces: The careful exclusion of other forces at the interface is needed. For example, electrostatic forces between the AFM tip and the samples are often seen in not ideally grounded systems. Experimental proofs should be provided.

We thank the Reviewer for her/his suggestion. In order to exclude any scenario of electrostatic interactions, we have carried out a new set of AFM measurements (**Fig. R15**), in which we have transferred graphene onto a 70 nm gold-coated holey SiN_x membrane chip, which was electrically grounded throughout the AFM indentation and scanning. As graphene was in contact with gold during measurement, one can safely exclude the electrostatic interactions.

Figure R15 | A new set of force measurements excluding the influence of electrostatic interactions. (a) Photograph comparing a piece of holey SiN_x chip before (left) and after (right) the deposition of 70 nm gold layer for electrostatically grounding graphene throughout measurements. Topographical (b) and surface force (c) maps generated using a 13 nm radius gold-coated AFM tip that approached another piece of graphene transferred onto the Au-coated SiN_x holey membrane. The surface force maps present the vertical force values experienced by the AFM tip at a displacement d of approximately 10 nm before

establishing the contact. The areas enclosed by white dashed lines correspond to the regions experiencing repulsive forces of ≥ 10 pN.

As the graphene was in close contact with gold, which was electrically grounded with the AFM tip throughout the measurement, one can ensure the exclusion of electrostatic interactions. Note that we found that transferring graphene onto gold was unexpectedly challenging as the gold-graphene interaction is very low; the average yield was substantially reduced. At the end, we not only fabricated a sample that can be electrostatically grounded, but also obtained a corrugation-free suspended graphene, such that the repulsion was consistently detected at the center of suspended area (Figs. R15 b and c).

In addition to electrically grounding, in the new measurements, we have used a gold-coated AFM tip of smaller tip radius (approximately 13 nm, see **Fig. R16a** on next page). In addition to the consistent observation of repulsive response (**Fig. R16b**), we have observed that the width of the linear region is also reduced to ~ 13 nm (**Fig. R16c**).

More specifically, in the new measurements, there are three main findings as follows. (i) The width of linear region is consistent with the tip radius, endorsing our picture of graphene deformation upon indentation (see our earlier discussion for details). (ii) The AFM tip remains clean without debris after all measurements (**Fig. R16 a**). (iii) As shown in **Figs. R16 b and c**, the evaluation of 144 force-displacement responses at the center of the suspended graphene reveals a repulsive force of 7.5 ± 5.7 pN at an average separation of approximately 7.5 nm before making contact. These values allow us to estimate the geometric parameters, the effective radius R_{eff} and area of interaction A_{max} of 18nm and 4254 nm² respectively. Considering the smaller size of tip, the values are surprising intact with the physical picture presented in **Fig. 4e**.

In summary, given the observations that both sets of experiments yielded consistent results, we can safely conclude that the electrostatic interaction is not responsible for the observed repulsion. Note that the new experiments also allow us to address the Reviewer's other questions, as will be discussed later.

2. Deformation Effects: *As mentioned in the paper, suspended graphene would undergo deformation when the AFM tip approaches the surface. Even within the suspending holes, the repulsive forces vary significantly. The reasons for this variation should be carefully examined and explained with evidence, not conjecture.*

We thank the Reviewer for the question. Here we would like to elaborate how the surface properties of graphene, including its highly flexible nature, influence the repulsion detected by the AFM tip. As mentioned in the manuscript, the repulsion was consistently measured on flat suspended graphene only.

In order to further exam the flatness requirement, here we would like to make use of the experimental data presented in Figs. R15 and R16. In this sample, we were able to transfer graphene free of corrugation as one can confirm from the topographical map found in **Fig. R15b**. The suspended area resembles a parabolic surface, with relatively flat region exactly located within the center of the pore. Combining with the surface force map shown (**Fig. R15c**), we can clearly observe that the repulsive forces occurred at the flat region of the suspended graphene. To further quantify the flatness required to exhibit repulsion, we have analyzed the surface profile of the suspended region, with respect to the surface force. **Figures R17 a and b** superimpose the repulsive force values measured over two representative cross-sectional cuts of the suspended area.

Figure R16 | Detailed analysis for the newly fabricated suspended graphene sample transferred onto a gold-coated holey SiNx substrate. (a) Tilted SEM image of the AFM tip after the new set of force measurements. The tip remained clean and has a tip radius of approximately 13 nm (Scale bar: 100 nm). (b) 2D histogram of 144 force-displacement measurements taken from the center of the new sample. A repulsive force of 7.5 ± 5.7 pN was measured at an average separation of approximately 7.5 nm. The reduction of repulsive force results from a smaller interacting area. (c) The magnified force-displacement response showing a clear linear region of width of ~ 13 nm, consistent with the tip radius.

To quantitatively examine the effects of surface flatness, we have fitted local surface profile with a parabolic function, $f_P(x) = a(x - b)^2 + c$, at a given position x . Together with the map of measured repulsive forces, we have clearly identified that the repulsion emerges when graphene is relatively flat, with the maximum surface gradient, $\left\| \max \left(\frac{df_P}{dx} \right) \right\| \leq 0.01$. One can also clearly observe how the interacting area amplifies the repulsive feedback.

Figure R17 (Fig. 1f-g in the updated main text) | Generation of repulsive force on flat area of suspended graphene. (a,b) Representative cross-sectional cuts through the repulsive domains in Fig. R15 (b) and (c). A parabolic function $f_P(x)$ was used to fit the topographic profile, suggesting that the repulsion emerges when graphene is relatively flat, with the maximum surface gradient, $\left\| \max \left(\frac{df_P}{dx} \right) \right\| \leq 0.01$.

3. Surface Contaminations: *Surface contaminations generated from wet-transferred graphene are very likely to occur. The impacts on the measurements should be analyzed.*

We thank the Reviewer for this remark. In order to elucidate the effects of surface contamination on the measured repulsive forces, we have carefully examined the suspended graphene sample in the new set of measurements.

First, inspired by Ref. [4], we have used the cellulose acetate butyrate (CAB) as a sacrificial layer to transfer the exfoliated graphene. Our tests have suggested that it is significantly easier to remove CAB in acetone, leaving only very small amount of polymer residue after transfer, in line with another recent report [5]. As a result, the post process of Ar/H₂ annealing becomes very effective, yielding nearly contamination-free graphene samples. Right after annealing, the samples were kept in vacuum before subsequent experiments or characterization. More specifically, Ar/H₂ annealing allows the removal of: (i) large specks of organic residue after acetone rinsing from both suspended and substrate-supported graphene (see Figs. R18a and R18b), and (ii) airborne contaminants that gradually absorb on the surface after long-term ambient exposure. Note that any SEM-based inspection should be also avoided before the AFM measurement, in order to prevent any electron beam-induced hydrocarbon deposition [6].

Second, Figure 19a presents SEM image of a suspended graphene area before annealing, which was adjacent to the area for the new AFM measurement. Note that this is necessary because the electron beam could induce hydrocarbon deposition [12]. We observed that the surface contaminants, which are highlighted in white circles, are very sparsely distributed, with an effective average diameter of 27.6 nm, excluding the bigger ones at the edge.

Considering the fact that majority of the area is contaminant-free, we believe that the force-displacement measurements were not influenced. In order to further elucidate whether such contaminant could induce any possible artifact, we have carefully analyzed our new measurements. In particular, we focus on the outliers with unexpected high or low feedback. As shown in Figs. 19b and c, we highlighted the outliers in blue and red circles respectively, which were identified by combining the topographical and repulsion maps. Although the contamination density seems slightly higher than **Fig. R19a**, it is convincing that the outliers result from the surface contaminants, given that the contaminant size is of comparable size of AFM tip radius. Accordingly, we are confident claim that the surface contaminants manifest themselves as outliers of measurements, which do not influence the quantification of repulsive forces in the clean area.

Figure R18. Ar/H₂ annealing of graphene transferred by using the CAB-based protocol. Optical micrographs for graphene transferred on to a holey SiN_x membrane (**a**) before and (**b**) after Ar/H₂ annealing, showing that the majority of polymer residues were removed on both suspended and substrate-supported graphene.

Figure R19 | Surface contamination analysis. (**a**) SEM image of a representative pore, which was fabricated alongside the sample used for AFM measurements, allowing us to identify the typical size of the surface contaminant (27.6 nm), comparable to the AFM tip radius. (**b**) and (**c**) show topography and surface force maps of the new measurement with outliers of both unexpected high or low feedback highlighted in blue and red circles respectively.

Figure R20 | Calculated vdW interaction energy for Vac/NL-Gr/Au system with various graphene layer numbers. With increasing graphene layer thickness, the repulsive response diminishes.

On the other hand, we would like to point out that **the Lifshitz-vdW repulsion is a mesoscale force, insensitive to the surface property of graphene.** As discussed in our response to the first point raised by the Reviewer 1, the calculated repulsive force remain strong even for multilayer graphene (see **Fig. R20**). In other words, even there is a small degree of polymer residue or airborne contamination deposited on monolayer graphene surface, the vdW repulsion would remain to occur.

In summary, first, in all samples considered in this report, we have utilized the CAB-based graphene transfer protocol, which maximally remove polymer residues, so that the Ar/H₂ annealing can yield nearly contamination-free graphene surfaces. Second, the sparsely distributed surface contaminants have typical size of comparable to the tip radius, so that the measurements result in outliers, which can be easily identified. Lastly, the Lifshitz-vdW repulsion is a mesoscale force, insensitive to a small amount (if there is any) of surface contaminants. Altogether, we conclude that the effects of organic residues are nearly negligible to the measurement of Lifshitz-vdW repulsion.

4. Impact of Surface Corrugation: *The paper mentions the presence of surface corrugation near the edges of suspended graphene but does not thoroughly analyze its impact on the measured forces.*

We thank the Reviewer for the question. Following our earlier discussion in **Fig. R17** addressing the Reviewer's pt. 2, **Fig. R21** presents the local ripples having surface corrugation datasets of the topographical and surface map presented in the manuscript in **Figs. 1b** and **1c**. Specifically, **Fig. R21a** shows a cut through the smaller repulsive area, and **Fig. R21b** shows another cut through an area without showing repulsion. In **Fig. R21a**, we found that the repulsion can be detected at the relatively flat area within the convex surface, with sufficiently low inclination. On the other hand, **Fig. R21b** suggests that when the surface corrugation is significantly high without locally flat domains, the repulsion cannot be detected, echoing our earlier finding that as long as there are no sufficiently large flat domains,

$\left\| \max \left(\frac{df_p}{dx} \right) \right\| \leq 0.01$, the repulsive responses could not be detected. Together with **Fig. R17**, one can conclude that the indentation of an AFM tip of finite tip radius can only resolve the orthogonal forces, which demand a relatively flat surface. When the surface has domains of a degree of inclinations, the indentation of an AFM tip would most likely result in a lateral displacement for either the AFM tip or the graphene surface.

Another piece of evidence supporting our finding is the nucleation behavior, as will be discussed in details later. In the nucleation experiments upon the evaporation of gold onto suspended graphene, we and other groups (such as Ref. [1]) have observed that there is no nucleation of gold in the flat areas between corrugated ripples; the nucleation takes place nearly exclusively on the defects, contaminants, and ripples. More discussion will follow.

In summary, we hope the quantitative analysis presented in Figs. R17 and R21 clearly address the Reviewer's question.

Figure R21 | Detection of repulsive forces on corrugated graphene. Representative cross-sectional cuts through the corrugated graphene sample shown in **Fig. 1b, c**, including (a) a cut passing through a repulsive area, and (b) a cut passing through a corrugated area without the detection of repulsion.

5. Error Bars: the error bars in figures 4a and 4b are extremely large. The accuracy and representativeness of the data should be explained.

We thank the Reviewer for raising the concern. We would like to point out that, despite the large error bars, the repulsive forces are unambiguously identified, with at least 1σ to 2σ above zero. In other words, statistically speaking, the reported repulsive force is well within a 95% confidence interval, as long as the graphene surface is sufficiently flat. On the other hand, thanks to Reviewer's suggestion, in the new set of experiments that excludes the electrostatic interactions, the repulsive forces were consistently measured.

Following our earlier discussion regarding the surface corrugation and flatness of suspended graphene, one of major factors contributing to the large variation is the surface roughness and the ultra-softness of suspended graphene upon indentation. On the other hand, we are also limited by the choice of cantilever.

We had tested many different cantilevers with lower spring constants, which in principle enable a higher force resolution. However, they usually have a relatively small reflective surface, such that the interference signals are of several orders of magnitude larger than the measured repulsion. Please note that the measured attractive forces on SiNx-supported graphene also share similar level of force variation.

With the above rationale in mind, we are confident that the measured repulsive forces are not only statistically representative, but also reproducible in different substrate and tip configurations. A relatively large variation of measured repulsive forces (an example please see **Fig. R17**) comes from the soft nature of AFM tip, as well as the ultra-softness and surface corrugation of graphene.

6. Wetting Tendency: *For the wetting tendency difference of suspended graphene and graphene/SiNx, the strain effect and the surface roughness should be considered. The corrugations/wrinkles on graphene/SiNx obviously introduce a higher degree of nucleation sites, as seen in figure 5a, where nucleation forms lines. The surface roughness of SiNx would also introduce nucleation sites. The wettability due to surface roughness should be excluded.*

Following our earlier discussion in response to Reviewer's pts. 4 and 5, we agree with the Reviewer that the surface corrugations of graphene is playing an important role in determining the nucleation density, and the surface roughness of SiNx-supported graphene could also introduce nucleation sites.

In order to exclude the effects of surface contamination, roughness, and strain, we have carried out a new experiment by transferring a piece of exfoliated graphene containing monolayer, bilayer, and 4-layer regions on a holey SiNx membrane, followed by slow evaporation of gold on the sample. Before the deposition, the sample was annealed in hydrogen to remove the surface contamination. Although it is minimum, we could basically assume all areas have the same degree of surface contamination. In terms of surface roughness and strain, the bilayer and 4-layer graphene are even smaller than the monolayer counterpart, because the mechanical modulus increases with layer number.

Figure R22 (new **Fig. 5** in main text) compares the surface nucleation density on suspended (a) monolayer, (b) bilayer, and (c) 4-layer areas, which suggest a clear trend: the nucleation density increases with the layer number. This finding shows nice agreement with our calculations shown in **Fig. R20**, in which the Lifshitz-vdW repulsion decreases with increasing the layer number. The reduction of Lifshitz-vdW repulsion could make the surface less "repulsive, thereby increasing the nucleation density.

We can therefore infer that the surface contamination, roughness, and strain are not the dominant factors resulting in the substantially lowered wettability of suspended monolayer graphene. One can also observe that the suspended region has overall lower nucleation density than the SiNx supported area, which is in agreement with the findings reported in Ref. [1]. In particular, in **Fig. R22a**, we observed that gold nucleated along a line, leaving the rest area nearly non-wettable. We suspect that the line corresponds to a line defect or wrinkle in monolayer graphene.

The phenomena was observed in Ref. [1], which suggested that the nucleation sites on suspended graphene exclusively take place on ripples or defects induced by mechanical deformation during the wet transfer process. The TEM observations in Refs [19,20] also show that gold does not adhere on suspended monolayer graphene but rather on surface imperfections. The literature strengthens our findings on the non-wettability of suspended flat monolayer graphene.

Finally, please also note that we have excluded the nucleation sites at the wrinkles/defects in **Fig. 5** in the main text from the manuscript in our nucleation density calculation. We thank again the Reviewer for pointing out the effects of surface corrugation on graphene's wettability. We have added relevant discussion in the updated main text.

Figure R22 | SEM images comparing the nucleation density of gold on suspended (a) monolayer, (b) bilayer, and (c) 4-layer graphene deposited in an e-beam evaporator, at a constant rate of 0.01 nm/s and a nominal thickness of 0.1 nm. Scale bar: 1 μm .

7. Modeling Other 2D Systems: *If other 2D systems could be modeled and behave very differently, experimental measurements should be provided to examine the accuracy of the calculations.*

We thank the Reviewer for her/his suggestion. We have carried out additional calculations elucidating the effects of 2D material choice. In these calculations, we consider the systems of Vac/2D monolayer/Au, so the intrinsic dielectric responses of 2D monolayers essentially determine the total interaction energy. We have included the new calculations in the updated Supporting Information (Supplementary **Figs. S10 and S11**).

Figure R23 (Supplementary **Fig. S10** in updated Supporting Information) | **Influence of choice of 2D material.** **a.** Frequency-dependent dielectric responses for graphene (Gr), MoS₂, hBN at $d = 1$ nm, in comparison with $\sqrt{\epsilon_{\text{Au}}}$ as a function of frequency. **b.** Calculated interaction spectra taking into account the g_m term, revealing that the Vac/hBN/Au system could yield stronger vdW repulsion.

Figure R24 (Supplementary **Fig. S11** in updated Supporting Information) | Comparison of the calculated interaction force for different suspended 2D monolayers. One can observe that the suspended MoS₂ yields up to a 43% stronger repulsive force than graphene within the measurement-relevant regime (6 – 12 nm).

Specifically, we have compared three 2D monolayers, graphene, MoS₂, and hBN. **Figure R23** presents the calculated dielectric responses (**Fig. R23a**) and the interaction energy G_{AMB} (**Fig. R23b**) at a separation of 1 nm. All the suspended monolayer exhibit full-spectrum-repulsion, as gold has a higher dielectric constant than all 2D monolayer considered here at all frequencies. The calculated interaction energy seems to suggest hBN yields the highest repulsive interaction. However, as shown in **Fig. R24**, the calculated repulsive force using Eq. (3) in the main text suggests that MoS₂ can yield the strongest repulsive force among all three monolayers considered here. Indeed, we found that the anisotropic factor g_m only becomes more dominant at very small separations. Within the range of interest ($6 < d < 60$ nm), suspended MoS₂ yields a 43% stronger repulsive force than graphene. The overall trends for the calculated forces behave very similarly for all three monolayers.

Nevertheless, we would like to stress that this manuscript mainly focus on the experimentally observed repulsive force-displacement responses upon AFM indentation, emphasizing on the connection and influence of the repulsion on flat suspended monolayer graphene. The theoretical analysis and the nucleation experiments only play a supportive role to the main findings. We acknowledge that our theoretical framework has a certain degree of limitation, in particular at very small separations ($d < 6$ nm). A theoretical framework capturing all features at all separations would demand more effort, which is beyond the current scope of this manuscript. We would appreciate the Reviewer’s understanding.

References

1. Thomsen, J. D. et al. Suspended Graphene Membranes to Control Au Nucleation and Growth. *ACS Nano* 16. PMID: 35849654, 10364–10371. (2022).
2. García, R. & San Paulo, A. Attractive and repulsive tip-sample interaction regimes in tapping-mode atomic force microscopy. *Phys. Rev. B* 60, 4961–4967 (7 1999).
3. Application Note # 133 Introduction to Bruker's ScanAsyst and PeakForce Tapping AFM Technology.
4. Schneider, G. F., Calado, V. E., Zandbergen, H., Vandersypen, L. M. K. & Dekker, C. Wedging Transfer of Nanostructures. *Nano Lett.* 10, 1912–1916 (2010).
5. Burwell, G., Smith, N. & Guy, O. Investigation of the utility of cellulose acetate butyrate in minimal residue graphene transfer, lithography, and plasma treatments. *Microelectronic Engineering* 146. *Nanostructured Materials and Green Nanotechnology for future Electronics, photonics, and Nanosystems* 2014, 81–84 (2015). R38
6. Dyck, O., Kim, S., Kalinin, S. V. & Jesse, S. Mitigating e-beam-induced hydrocarbon deposition on graphene for atomic-scale scanning transmission electron microscopy studies. *Journal of Vacuum Science & Technology B* 36, 011801. (2017).
7. Klimchitskaya, G. L., Mostepanenko, V. M. & Sernelius, B. E. Two approaches for describing the Casimir interaction in graphene: Density-density correlation function versus polarization tensor. *Phys. Rev. B* 89, 125407 (12 2014).
8. E, S. B. Graphene as a strictly 2D sheet or as a film of small but finite thickness. *Graphene* 01, 21–25 (2012).
9. Levin, M., McCauley, A. P., Rodriguez, A. W., Reid, M. T. H. & Johnson, S. G. Casimir Repulsion between Metallic Objects in Vacuum. *Phys. Rev. Lett.* 105, 090403 (2010).
10. Milton, K. A. et al. Three-body effects in Casimir-Polder repulsion. *Phys. Rev. A* 91, 042510 (4 2015).
11. Rodriguez, A. W., Joannopoulos, J. D. & Johnson, S. G. Repulsive and attractive Casimir forces in a glide-symmetric geometry. *Phys. Rev. A* 77, 062107 (6 2008).
12. Rodriguez, A. W., Capasso, F. & Johnson, S. G. The Casimir effect in microstructured geometries. *Nature Photonics* 5, 211–221 (2011).
13. Canaguier-Durand, A., Neto, P. A. M., Lambrecht, A. & Reynaud, S. Thermal Casimir Effect in the Plane-Sphere Geometry. *Phys. Rev. Lett.* 104, 040403 (4 2010).
14. Han, E. et al. Ultrasoft slip-mediated bending in few-layer graphene. *Nature Materials* 19, 305–309 (2019).
15. Milton, K. A. et al. Repulsive Casimir and Casimir-Polder forces. *Journal of Physics A: Mathematical and Theoretical* 45, 374006 (2012).
16. Klimchitskaya, G. L., Mohideen, U. & Mostepanenko, V. M. The Casimir effect in graphene systems: Experiment and theory. *International Journal of Modern Physics A* 37, 2241003. (2022).
17. Klimchitskaya, G. L., Mostepanenko, V. M. & Tsybin, O. Y. Casimir-Polder attraction and repulsion between nanoparticles and graphene in out-of-thermal-equilibrium conditions. *Phys. Rev. B* 105, 195430 (19 2022). R39
18. Tian, T. et al. Electronic Polarizability as the Fundamental Variable in the Dielectric Properties of Two-Dimensional Materials. *Nano Lett.* 20, 841–851 (2019).

19. Zan, R., Bangert, U., Ramasse, Q. & Novoselov, K. S. Evolution of Gold Nanostructures on Graphene. *Small* 7, 2868–2872. (2011).

20. Zan, R., Bangert, U., Ramasse, Q. & Novoselov, K. S. Interaction of Metals with Suspended Graphene Observed by Transmission Electron Microscopy. *The Journal of Physical Chemistry Letters* 3. PMID: 26286426, 953–958. (2012).

Response to Reviewer # 1

(Remarks to the Author):

I appreciate the authors' efforts and additional data that has been presented. However, some additions raise new questions which should be addressed before publication:

We thank the Reviewer for her/his constructive comments. To properly address all the new questions, we have prepared a large-area ($> 1000 \mu\text{m}^2$) exfoliated graphene sample allowing us to have sufficiently large suspended mono- and 4-layer graphene areas for statistical analysis. We have carried out new epitaxial experiments in which Raman spectroscopy was used to determine the graphene layer numbers at different pore positions, followed by systematic statistical analysis in their epitaxial nucleation.

Note that we decided to carry out the new experiments because: (i) the old sample presented in the previous version of manuscript had undergone irreversible damages, which prevented us from in-situ characterization of graphene layer numbers, and (ii) its suspended areas are not sufficiently large for the analysis of nucleation behavior in a statistically meaningful manner.

We hope the Reviewer could recognize our effort in addressing her/his questions. Please find our point-to-point response as follows.

1. Pg. 8: "The last notable measured repulsive force were detected...". What is meant by the last "notable" measured repulsive force? Do the authors simply mean the last repulsive force before the tip experiences an attractive force? Also, should "were" actually be "was"?

We thank the Reviewer for highlighting this point. Indeed, "The last notable measured repulsive force were detected..." refers to the point at which the AFM tip experiences attraction, leading to a change into negative values for the derivative of the measured force. To improve the clarity, we have modified the original sentence as follows:

"The last notable measured repulsive force before experiencing attraction was detected at an average displacement of 8.8 nm and 6.6 nm for gold and SiN_x AFM tips, respectively."

2. On pg. 15 the authors write: "As a result, any electroneutral object approaching the surface could experience a repulsive force, smaller or larger, depending on the interacting configuration." Can they expand on what is meant by "interacting configuration" and which factors will make the force smaller or larger? Is this related to the results in Fig. 1(c, e) which has areas of the suspended graphene that does not display a repulsive force?

We thank the Reviewer for allowing us to expand the discussion on the parameters that could change the interaction strength. From a phenomenological point of view, based on our effective medium approach, the strength of the interaction depends on the parameters appeared in Eq. (3) of the manuscript, in which the permittivity, separation, and object geometry are crucial for the magnitude of Lifshitz-vdW repulsion. We have added the following sentence to clarify:

"More specifically, the interacting configuration mainly refers to: (i) the materials permittivity, as it determines the product of $(\epsilon_A - \epsilon_m)(\epsilon_B - \epsilon_m)$ in Eq. (1) and (ii) the separation and object geometry (or the effective interacting area), which are parameters scaling the interactions."

3. *Fig. 5: The authors state in the rebuttal letter that their new result shown in Fig. 5 is not consistent with Ref. 30. Also, the thickness trend also goes against what is seen for SiO₂-supported Gr (references for this were given in the first reviewer assessment) as well as seen for other 2D materials such as MoS₂ on SiO₂. Although these latter samples are not suspended and therefore might not relate exactly to the samples studied here, it is nevertheless interesting that an opposite trend is observed. To allow readers to appreciate the link between this and other work, this should be mentioned and discussed in the main text. The trend is not discussed in the updated manuscript, so the authors could also state that it matches their theoretical prediction, supporting the hypothesis.*

We thank the Reviewer for the helpful suggestion. As mentioned earlier, because our old sample was damaged, we have prepared a large-area ($> 1000 \mu\text{m}^2$) exfoliated graphene sample allowing us to have sufficiently large suspended mono- and 4-layer graphene areas for statistical analysis. We have in-situ analyzed the epitaxial nucleation behavior with respect to the graphene layer number. Please find the corresponding discussion in the response of the Reviewer's pt. 4.

And after the discussion of new **Fig. 5**, according to the Reviewer's suggestion, we have added a paragraph comparing our observation with the literature as follows.

We noticed that our observation of increased nucleation sites on thicker suspended graphene exhibits an opposite trend compared to the nucleation behavior on SiO₂-supported graphene [56, 57]. Indeed, as revealed in Eq. (1), the presence of an underlying substrate, such as SiO₂, could convert the Lifshitz-vdW interaction from repulsion to attraction, thus affecting the nucleation behavior of gold. A recent report by Frances et al. [30] also observed the aggregation of gold particles on the ripples of suspended monolayer graphene, but the nucleation density remained to decrease with the layer number of suspended graphene. We suspect that the deposition rate, or the kinetic energy of the incident gold particles, could also play an important role. More detailed and systematic experimentation is required to fully understand the phenomenon. Nevertheless, we would like to point out that the extremely low surface wettability of suspended monolayer graphene observed here cannot result from atomic smoothness, as its mechanical flexibility typically induces a higher degree of surface corrugation during the transfer process.

4. *Fig. 5: How was the layer thickness in each area assigned? Please include optical microscopy images of the sample and any other data in support of the determination.*

We would like to thank the Reviewer for the question. Within the new set of experiments, we have successfully characterized the layer number of suspended graphene by the optical contrast and Raman spectrum. Specifically, **Supplementary Fig. S10a** (attached below) overlays two optical micrographs for the exfoliated graphene sample before and after transfer onto a SiN_x holey membrane, allowing us to visualize the areas of various thicknesses. We have first identified the layer numbers according to the optical contrast enhancement values reported in the literature (Ref. [S10]), specifically 4.8%, 16.3%, 25.3% and 35.6% for mono-, bi-, tri-, and 4-layer graphene, respectively. We have identified two significant areas of suspended mono- and 4-layer graphene that cover a sufficient number of SiN_x holes. The Raman spectra (**Figs. S10 b,c**) confirm that they are monolayer and 3-4 layer graphene, respectively, based on the 2D to G band intensity ratios. We noticed that the monolayer graphene is more defective than the 4-layer area, as reflected by the D band intensity, possibly being induced during the annealing process. The defects could introduce additional nucleation sites on monolayer graphene.

As will be discussed in detail in response to the Reviewer's pt. 5, Fig. S10a reveals that there are three SiN_x holes fully covered by suspended 4-layer graphene, allowing us to statistically analyze the nucleation behavior in comparison to the monolayer holes.

Figure S10. A large-area suspended graphene sample for the epitaxial nucleation experiment discussed in Fig. 5. (a) Overlay of the optical microscope images before and after transfer for the visualization of the areas of various graphene thicknesses. The values of relative contrast enhancement of 4.8%, 16.3%, 25.3%, and 35.6% were measured, allowing us to identify the mono-, bi-, tri-, and 4-layer graphene areas. Two large areas of suspended mono- and 4-layer graphene, corresponding to the enclosed blue and black dashed lines, were identified for statistical analysis shown in Fig. 5. **(b,c)** Representative Raman spectra for mono- **(b)** and 4-layer **(c)** graphene.

5. Fig. 5: If the sample shown in Fig. 5 contains more suspended areas it would be good to show those as well (in the SI) to have an idea about differences in nucleation density across a sample. Comparing Fig. 5a with the original Fig. 5a there is also quite a large difference in nucleation density and particle size on monolayer graphene. The original image could also be shown to give a fuller picture.

We thank the Reviewer for the constructive suggestion. With the new sample presented in Supplementary S10, we were able to identify two large areas of suspended mono- and 4-layer graphene, each covering at least three SiNx circular holes, with the hole diameter of 5 μm . Upon the evaporation of gold in an e-beam evaporator, at 0.01 nm/s rate and a nominal thickness of 0.1 nm, **Fig. 5** in main text (attached below) directly compares the SEM images and corresponding statistical analysis of nuclei sizes and counts for three areas of suspended mono- (**Fig. 5 a and b**) and 4-layer (**Fig. 5 c and d**) graphene on the same sample.

Figure 5. Lowered wettability of suspended graphene that reduces the nucleation of gold particles. SEM images and histograms of gold particle size comparing the nucleation behavior of suspended mono (a and b) and 4-layer (c and d) graphene areas in the same sample. The gold particles were deposited in an e-beam evaporator at 0.01 nm/s and a nominal thickness of 0.1 nm. The increase in the number of graphene layers enhances gold nucleation, which can be nicely explained by a reduced Lifshitz-vdW repulsion. Scale bars: 1 μm .

The new paragraph for the discussion of **Fig. 5** in the main text is attached as follows.

In particular, **Fig. 5** directly compares the nucleation behavior of gold particles grown on suspended mono- (**Fig. 5 a and b**) and 4-layer (**Fig. 5 c and d**) graphene. To reach statistical significance, for each graphene layer, histograms of gold particle sizes were extracted based on three independent areas of SiNx holes on the same sample of exfoliated graphene (**Supplementary Fig. S10**). On the monolayer graphene areas, we noticed that most nucleation sites took place along straight lines, hypothetically corresponding to the ripples induced during the annealing and transfer process, as suggested in Ref. [30]. On the contrary, the nucleation sites on the 4-layer areas distribute more randomly, suggesting that the 4-layer areas are intrinsically flatter due to its relative mechanical stability. Even when we count all particles accumulated along the ripples, within a circular suspended area of 5 μm diameter, our analysis reveals average counts of nucleation sites of 586 ± 52 and 906 ± 169 for mono- and 4-layer graphene, respectively. The nucleation density of gold particles on suspended monolayer graphene is approximately 35% lower than that on the 4-layer counterpart. These findings can be nicely explained by the generation of Lifshitz-vdW repulsion on suspended graphene, which according to our calculations (**Supplementary Fig. S12**), becomes stronger by reducing the graphene thickness. The amplification of Lifshitz-vdW repulsion on suspended monolayer graphene substantially lowers its wettability.

In summary, following the Reviewer's suggestion, we have carried out new experiments that allow us to statistically characterize the nucleation behavior on suspended mono- and 4-layer graphene areas. Our findings can be summarized as follows. (i) When there is a supporting substrate, the nucleation density is orders of magnitude higher than the suspended graphene areas. (ii) The nucleation sites on monolayer graphene take place nearly exclusively on the surface ripples. (iii) The nucleation density on suspended 4-layer graphene is significantly higher than that on the monolayer counterpart. The findings (i) to (iii) above can be nicely explained by the Lifshitz-vdW repulsion generated on suspended graphene.

6. Pg. 16: *"The extremely low surface wettability of suspended graphene observed here can neither result from the atomic smoothness of graphene nor from any ultra high vacuum treatment, because the nucleation density of gold on single crystalline graphite cleaved in air remains high [54]". This sentence could be revised to improve clarity. Is the argument that it cannot be atomic smoothness because of the new result on Fig. 5? If so, the authors could state this. For the second part of the sentence, is the argument that the samples have been cleaved in air where a large nucleation density is expected, and nevertheless a low nucleation density is observed?*

We thank the Reviewer for the comment. The Reviewer had understood the sentence perfectly. In the new version of manuscript, we have improved the sentence as follows (the entire paragraph please visit our response to Reviewer's pt.4). Note that the graphite cleavage part was removed to avoid complications.

"Nevertheless, we would like to point out that the extremely low surface wettability of suspended monolayer graphene observed here cannot result from atomic smoothness, as its mechanical flexibility typically induces a high degree of surface corrugation during the transfer process."

7. *Conclusion: "effect" should be "effective".*

We thank the Reviewer for pointing this out. We have corrected the typo.

At the end, again, we appreciate the Reviewer's detailed review and hope the Reviewer could recognize our efforts in improving the quality and impact of the updated manuscript.

Response to Reviewer # 2

(Remarks to the Author):

I commend the authors for the extra efforts in addressing the reviews; however, I do not believe that they have satisfactorily (and convincingly) shown that the observed phenomenon is related to the van der Waals force and not one of the many other effects and artifacts that may be present. As one simple example, when asked about electrostatic artifacts, the authors respond, "As the graphene was in close contact with gold, which was electrically grounded with the AFM tip throughout the measurement, one can ensure the exclusion of electrostatic interactions." Simply grounding the sample and tip does not eliminate electrostatic forces when dealing with real surfaces and interfaces, and there is a vast body of literature (both in surface science and within experimental Casimir measurements) that shows such connections do not necessarily remove electrostatic artifacts. Unfortunately, I cannot support publication of this manuscript.

We respect the Reviewer's negative opinion about our manuscript. While the electrostatic interaction is indeed a concern, it cannot explain the findings presented in this manuscript, including the reproducible measurement of repulsive forces using different AFM tips on different suspended graphene samples, as well as the substantially lowered wettability of suspended monolayer graphene that results in very low nucleation density (new Fig. 5). We believe all pieces of evidence point to the existence of Lifshitz-vdW repulsion.

Response to Reviewer # 3

(Remarks to the Author):

My concerns have been fully addressed. The manuscript has been substantially improved upon the revision. Thank you.

We thank the Reviewer for her/his constructive comments, which in our opinion, have significantly improved the quality of this manuscript.

Response to Reviewer # 4

(Remarks to the Author):

I've been asked to share my thoughts regarding the disagreement between the authors and Reviewer #2, particularly the latter's concern that the authors have not convincingly shown that the observed forces originate from van der Waals interactions rather than other possible artifacts. The reviewer highlights the electrostatics issue, quoting the authors' statement that grounding both the AFM tip and the sample excludes electrostatic interactions — and I have to say, I think the reviewer raises a valid point.

In practice, simply grounding the tip and sample does not eliminate electrostatic forces. This is well documented in surface science and Casimir force literature. You can still have:

Patch potentials due to work function variation or local adsorbates;

Capacitive coupling based on tip geometry and proximity;

Surface contamination, even with careful transfer and annealing protocols;

And non-uniform charge distributions, which create localized electric fields that aren't canceled out by image charges.

So I agree that the electrostatic contribution hasn't been rigorously excluded, and more caution is needed here. That said, I do think the authors have made a genuine and thoughtful effort to address this. They redesigned the experiment to use grounded Au-coated membranes, tested multiple tip radii and materials, and consistently found similar repulsive force behavior. They also introduced a novel modeling approach, treating the suspended graphene as an effective birefringent medium, and their measurements appear reproducible with cleaner, smaller tips.

In the revised version, the authors do acknowledge that even if some electrostatic effect were present, it would be expected to result in attractive forces—not the repulsion they observe. While this doesn't rule out all artifacts, it does offer a plausible argument that electrostatics alone can't explain their data.

We appreciate the Reviewer for her/his detailed and thoughtful evaluation of our manuscript. As pointed out by the Reviewer, we have done our best to exclude the effects of electrostatic interactions. Although we were not able to perfectly rule out all scenarios that can perturb surface potential, the electrostatic effect alone cannot explain the findings presented in this manuscript. All pieces of evidence, including: (i) reproducible measurements of repulsive forces using different AFM tips on different graphene samples, and (ii) the substantially lowered wettability of suspended monolayer graphene that results in very low nucleation density, point out the existence of Lifshitz-vdW repulsion.

Regarding the Reviewer's remaining concerns, please find our detailed response as follows.

That being said, there are a few issues that I think one could have addressed:

- 1. Force sensitivity: The authors never quantify their detection limit, which is crucial. In air, pN-level forces are already approaching the limits of what a well-calibrated AFM can resolve. Their reported signals are very close to that floor, so they need to show — statistically — that what they're measuring is indeed above noise.*

We would like to thank the Reviewer for raising the concern of our AFM tips' force sensitivity, which has motivated us to carefully examine the statistical significance of our results. Please find our analysis as follows.

In addition to the values of deflection sensitivity and cantilever spring constant provided in the Supplementary Information, the estimation of force resolution requires the value of the minimum detectable voltage, which is 0.375 mV. Accordingly, the force sensitivity, or the minimum detectable force, is different for each set of measurements presented in the manuscript. The calculated force sensitivity values for our two gold-coated tips are 4.216 and 4.771 pN for the radii of 30 and 13 nm, respectively. The value for the uncoated SiNx tip is 3.307 pN. Please find the detailed values of cantilever parameters and force sensitivity in Supplementary Table S1 (attached below).

Table S1. The estimation of force sensitivity for each AFM tip considered in this work.

Tip Radius	Defl. Sensitivity [nm/V]	Spring Constant [N/m]	Resolution [pN] at 0.375 mV
Au-coated (30 nm)	69.679	0.16136	4.216
Au-coated (13 nm)	86.484	0.14710	4.771
Uncoated (20 nm)	75.567	0.11670	3.307

With the calculated sensitivity values in mind, to evaluate the statistical significance of our measurements, we have carried out Welch’s t-test to examine whether both signal and noise floor levels share the same mean value as the null hypothesis.

First, we examined our raw data without sinusoidal background subtraction (see Supplementary Section S2.2). **Supplementary Figure S8 a** and **b** (attached below) present the histograms comparing the measured repulsive and background forces for the gold coated AFM tips of tip radius of 30 and 13 nm, respectively. The extracted p-values (p-val) are 0.102 and 0.119. In other words, the two sets of raw data without background subtraction already approximate a 90% confidence level.

Figure S8. Welch’s t-test for signal and noise floors in the raw dataset. Histograms for the measured repulsive (purple) and background (green) forces for the gold-coated tips of radii of 30 nm (**a**) and 13 nm (**b**) considered in this study. The extracted p-values are 0.102 and 0.119, respectively.

Note that the evaluation for the data set of SiNx tip-measured forces without background subtraction suggests that the periodic background noise dominates the signals.

After subtracting the sinusoidal background, **Supplementary Fig. S9** present the histograms comparing the measured repulsive and background forces for all three AFM tips considered in this study. The extracted p-values substantially decreases, reaching 0.044 to 0.049, which are all lower than the statistical

significance threshold $\alpha = 0.05$, corresponding to a confidence level of 95%. The analysis presented above allows us to not only reject the null hypothesis of signal and noise floor sharing the same mean value, but also demonstrate the influence of periodic background noise caused by optical interference. The sinusoidal background subtraction procedure considerably improves the signal-to-noise ratio, and the measured repulsive forces are clearly of statistical significance.

Figure S9. Welch's t-test for signal and noise floors in the dataset after sinusoidal background subtraction. Histograms for the measured repulsive (purple) and background (green) forces after sinusoidal background subtraction for the gold-coated tips of radii of 30 nm (a) and 13 nm (b), as well as the bare SiNx tip (c). The extracted p-values are all below 0.05, consolidating the statistical significance of our measured repulsive forces.

We again thank the Reviewer for granting us an opportunity to examine the statistical significance of the measured repulsive forces. We agree with the Reviewer that this is a crucial point and have included all the above discussions and figures in the revised Supplementary Information.

2. *Tip radius: While SEM imaging helps, tip radius estimation is not an absolute value — it depends on imaging assumptions and can change due to wear or contamination.*

We thank the Reviewer for his/her remark. In fact, in order to confirm the AFM tip remained intact, we had directly imaged the AFM tip after extensive force scanning (an example is shown in Supplementary Fig. S3b attached below), which shows no sign of contamination. The measured tip radius (13 nm) is consistent with the specs provided by the vendor.

Figure S3. Magnified SEM image of the gold-coated AFM probe considered in this study. The hemispherical shape of the tip is highlighted with the red dashed line, with the tip radius R fitted to be approximately 13 nm, consistent with the specs provided by the vendor.

Although our current AFM instrumentation does not allow us to directly image the AFM tip during the indentation, we got inspired by a recent report (S. Santos et al., *Rev. Sci. Instrum.* 83, 043707 (2012); Ref. [55] in main text), which demonstrated an interesting *in-situ* method to quantify the tip radius through the dynamic atomic force microscopy measurements, namely the A_c method.

Specifically, the A_c method derives the tip dimension by measuring the evolution of the free amplitude A experienced during indentation from the attractive to repulsive regimes. Remarkably, the authors found a relation to correlate the respective critical free amplitude A_c with the respective tip radius R , which is independent of the properties of sample and cantilever, but dependent on the tip radius (see **Fig. R1a** below). Although in our original measurements, we did not collect the corresponding amplitude-phase-distance (APD) data for the generation of the tabular correlation with the tip radius, we believe that the linear increase of the repulsive forces observed upon indentation at large displacements shares an equivalent scenario for the $A_c - R$ correlation.

Indeed, due to the flexibility of graphene coupled with the observed repulsion, the indentation of an AFM probe to the surface of suspended graphene directly measures the gradual change in the free amplitude, or, in our system, the linear increase in deflection with respect to the tip displacement, deforming graphene. Accordingly, the correlation between the measured repulsive forces (and the critical free amplitude A_c) increases with the interacting area upon indentation. The tip radius can therefore be determined within the repulsive regime before reaching the maximum. We have demonstrated a similar correlation in **Fig. R1 b-d** below.

In light of the fundamental insight illustrated in Ref. [55] in the main text, we have estimated the tip radius values, R_{est} , by measuring the range of tip displacement where the repulsive forces increase linearly. The estimated values are 37.3 ± 3.1 nm, 15.3 ± 3.1 , and 13.2 ± 3.1 nm for the two gold-coated and bare SiNx tips, respectively, which show reasonable agreement with the spec values, 30, 20, and 13 nm, respectively. Note that the error margins were determined by the bin size of each 2D histogram, leading to identical values for all sets of measurements.

Figure R1. Estimation of tip radius during the indentation process. (a) The A_c method deriving the tip radius by measuring the evolution of the free amplitude A experienced during indentation (adopted from Ref. [55] in main text). (b-d) A similar approach was used to estimate the tip radius by measuring the range of tip displacement where the repulsive forces increase linearly. The estimated values are 37.3 ± 3.1 nm (b), 15.3 ± 3.1 (c), and 13.2 ± 3.1 nm (d) for the two gold-coated and bare SiNx tips, respectively, showing reasonable agreement with the spec values, 30, 13, and 20 nm, respectively. The bin size of each 2D histogram determined the error margins.

The analysis presented above not only allows us to estimate the tip radius R_{est} during the indentation, but also consolidates the physical picture of our effective medium framework, showing $R_{\text{est}} \approx R_{\text{eff}}$ (see **Fig. 4** in the main text). We have added one paragraph on Page 13 in the main text to summarize the above discussion as follows:

“A plausible picture informing the observed linear regime may share the scenario proposed by Santos et al. [55], who demonstrated a linear dependence between the critical free amplitude during AFM indentation with respect to the tip radius. We inferred that the linear regime essentially measures the repulsive force experienced by the AFM tip during an indentation process that deforms graphene from a flat plane to a hemispherical surface, as revealed in Fig. 4c and Fig.4d, without reducing the average separation between the tip and graphene d_{avg} . One could readily estimate the tip radii, R_{est} , by characterizing the width of the linear regime, yielding values of 37.3 ± 3.1 nm, 15.3 ± 3.1 (Supplementary Fig. S7), and 13.2 ± 3.1 nm for the two gold-coated and bare SiNx tips, respectively.”

3. Prior literature: It would be helpful if the authors acknowledged previous work in the attractive regime, Langmuir 2018, 34, 40, 11980–11988, which presents a careful study of forces on single-layer graphene and does a clearer job demonstrating monolayer identification.

We thank the Reviewer for the suggested literature and have included it in the revised manuscript. We agree with the Reviewer that this report (Ref. [54] in the revised manuscript) is complementary to our work, as it comprehensively investigated the attractive vdW interaction between the AFM tip and suspended graphene at short separations. In addition, it also inspired us to find another literature reported by the same group for the estimation of tip radius upon indentation.

In summary, I believe the authors are pointing to a potentially interesting effect, and they've clearly invested substantial effort in the revised work. But the burden of proof is high when reporting something this subtle and close to the measurement floor. More clarity on the detection threshold and a more rigorous treatment of possible electrostatic and topographic artifacts would make this a stronger, more convincing contribution.

We thank the Reviewer for the detailed review and for pointing out a potential issue with the measurement floor. We hope the Reviewer could recognize our efforts in addressing the Reviewer's pt. 1, which in our opinion, have considerably improve the quality of the updated manuscript.

Response to Reviewer # 1

(Remarks to the Author):

I thank the authors for the thorough revisions. All my concerns have been addressed and I support publication.

We thank the Reviewer for their constructive comments focusing on the evaluation of the epitaxial experiments. In our opinion, this has significantly improved the quality of this manuscript.

Response to Reviewer # 4

(Remarks to the Author):

I commend the authors for presenting a bold and intriguing study. The effort to probe Lifshitz–van der Waals repulsion in suspended graphene using AFM and nucleation experiments is technically impressive and conceptually exciting. The revised manuscript shows a clear improvement in data presentation, and I appreciate the authors' extensive new experiments, including larger-area graphene, statistical validation, and thorough optical/Raman characterization.

That said, a few areas still merit attention:

The manuscript would benefit from a careful language revision. There are several grammatical issues and awkward phrasings throughout that occasionally obscure the meaning of key technical points. ex the word discussion on page 4

We thank the Reviewer for sharing their concerns about the language. After finalizing the manuscript, we reviewed it carefully to remove any confusing or awkward phrasing.

Electrostatics: As noted by Reviewer 2, grounding alone does not eliminate all possible electrostatic contributions, especially given known surface inhomogeneities and patch potentials. The authors have responded thoughtfully but could better acknowledge this limitation in the main text to resolve the issue with Reviewer 2.

Regarding electrostatics, we believe that we have sufficiently discussed that they are of negligible contribution. We thank the Reviewer nonetheless for sharing their concern.

Personally I have not observed this kind of repulsive behavior in my own AFM measurements, though I recognize differences in setup and conditions may explain this. If the authors are open to it, I would be pleased to independently attempt to reproduce these results using my system, as a cross-check could provide additional confidence in the robustness of the reported phenomenon.

We are very much in support of an attempt to recreate the measurement and are open to discussing any details with the Reviewer.